# OmniFace: Bridging the Image-to-Video Gap for High-Fidelity Face Swapping via Diffusion Transformer

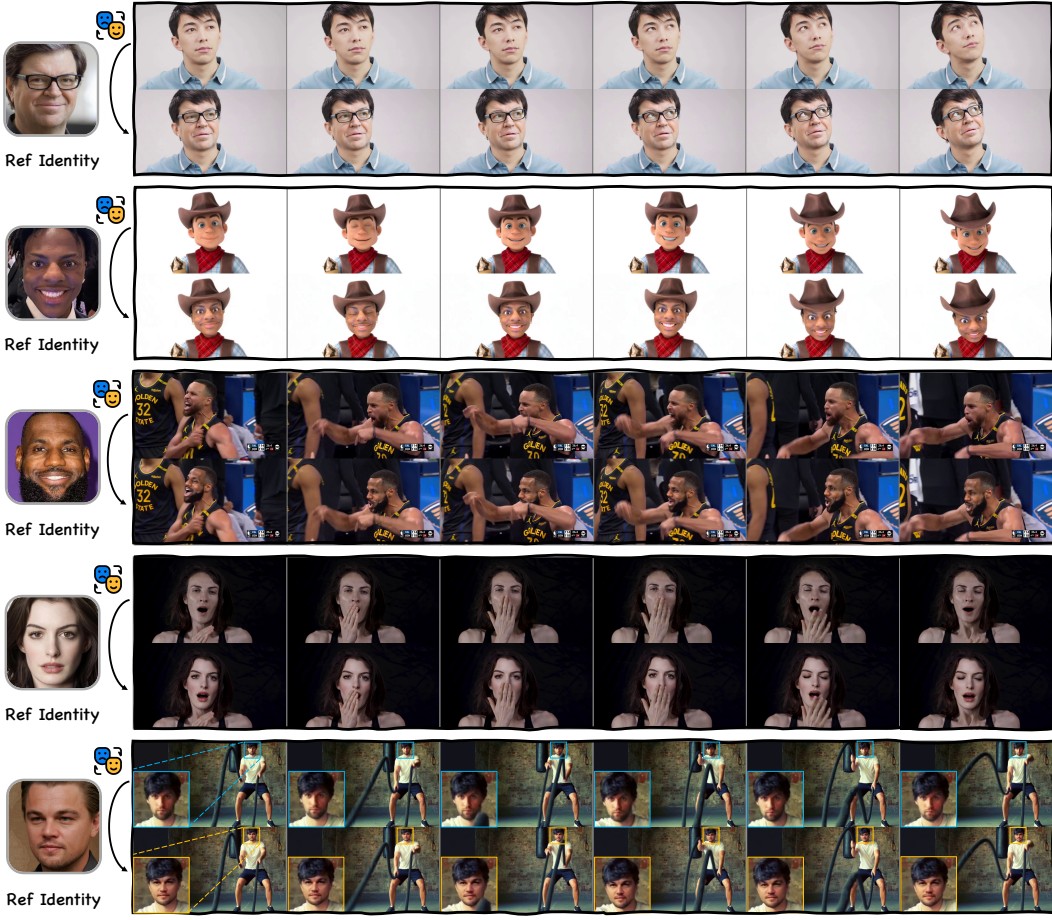

Figure 1: **Showcase of *OmniFace*.** *OmniFace* robustly handles challenging scenarios, e.g., complex expressions, animation, large angles, occlusions, and small faces.

## Abstract

Video Face Swapping (VFS) requires seamlessly injecting a source identity into a target video while meticulously preserving the original pose, expression, lighting, background, and dynamic information. Existing methods struggle to maintain identity similarity and attribute preservation while preserving temporal consistency. To address the challenge, we propose a comprehensive framework to seamlessly transfer the superiority of Image Face Swapping (IFS) to the video domain. We first introduce a novel data pipeline *SyncID-Pipe* that pre-trains an Identity-Anchored Video Synthesizer and combines it with IFS models to construct bidirectional ID quadruplets for explicit supervision. Building upon paired data, we propose the first Diffusion Transformer-based framework *OmniFace*, employing a core Modality-Aware Conditioning module to discriminatively inject multi-model conditions. Meanwhile, we propose a Synthetic-to-Real Curriculum mechanism

and an Identity-Coherence Reinforcement Learning strategy to enhance visual realism and identity consistency under challenging scenarios. To address the issue of limited benchmarks, we introduce **IDBench-V**, a comprehensive benchmark encompassing diverse scenes. Extensive experiments demonstrate *OmniFace* outperforms state-of-the-art methods and further exhibits exceptional versatility, which can be seamlessly adapted to various swap-related tasks.

# 1 INTRODUCTION

Face swapping aims to generate an image or video that combines the identity of a source face with the attributes (such as background, pose, expression, lighting) from a target image or video. This technique has sparked considerable research interest, due to its significant potential for practical applications in film production, creative design and privacy protection. Unlike Image Face Swapping (IFS), Video Face Swapping (VFS) presents more challenges, as it introduces additional critical constraints on temporal identity continuity, pose consistency, and environment preservation. Existing studies on Image Face Swapping (IFS), such as Ye et al. (2025); Han et al. (2024), have achieved remarkable success in maintaining identity similarity and preserving attributes. However, directly applying these IFS methods frame-by-frame to video sequences often leads to significant challenges in temporal consistency, resulting in noticeable flickering and jittering artifacts. Recently, the rapid advancement of diffusion-based video generation models has greatly propelled the development of Video Face Swapping (VFS). While methods like VividFace Shao et al. (2024), DynamicFace Wang et al. (2025), HiFiVFS Chen et al. (2024), and CanonSwap Luo et al. (2025) have improved the coherence and generation quality of VFS, their capabilities in terms of identity similarity and attribute preservation still lag behind those of state-of-the-art IFS models. The fundamental difference between IFS and VFS lies in the dynamic nature of video, which requires consistent preservation of motion and expression across frames. This observation inspires us to explore whether we can bridge the gap between image and video domains by supplementing these dynamic signals, thereby harnessing the strengths of IFS to significantly boost VFS performance.

Building on these insights, we propose a comprehensive framework comprising a novel data pipeline and a customized architecture to enhance VFS performance significantly. Our proposed data curation pipeline, **SyncID-Pipe**, seamlessly transfers the superiority of IFS to the video domain. Specifically, the pipeline pretrains a pose-driven First-Last-Frame video generation model, which we term the Identity-Anchored Video Synthesizer (IVS). The IVS employs an adaptive pose attention mechanism to inject pose information into First-Last-Frame video foundation models. This enables the model to generate a video consistent with the content of the given start and end frames and the actions specified by the pose sequence. Subsequently, we combine it with IFS models to construct bidirectional ID quadruplets for explicit supervision. Furthermore, to enhance data reliability, we propose an expression adaptation strategy to achieve effective expression transfer, incorporating an enhanced background recomposition mechanism to ensure strict environment alignment in the paired videos.

Building upon paired data, we develop **OmniFace**, the first video face swapping framework based on Diffusion Transformer (DiT) models Peebles & Xie (2023a), achieving high-similarity and superior-coherence results. We first introduce a Modality-Aware Conditioning (MC) mechanism that discriminatively injects conditions from multiple modalities, enabling condition decoupling and feature fusion. Furthermore, we design a novel Synthetic-to-Real Curriculum learning strategy to strengthen visual realism while maintaining identity similarity. To further enhance the preservation of facial dynamics under challenging scenarios, we develop an Identity-Coherence Reinforcement Learning (IRL) mechanism, which significantly improves model robustness in complex motions.

Through the aforementioned improvements, our approach enables effective video face swapping across diverse scenarios, as illustrated in Fig. 1. Due to the limited video-face-swapping benchmarks, we introduce **IDBench-V**, a comprehensive benchmark encompassing a wide spectrum of videos with varying head poses, facial expressions, and lighting conditions. We conduct extensive evaluations on *IDBench-V*, demonstrating that *OmniFace* achieves clear advantages over state-of-the-art methods, both quantitatively and qualitatively. Notably, our proposed framework exhibits exceptional versatility and can be seamlessly adapted to various swap-related tasks. Overall, our contributions are summarized as follows.

**Technology.** 1) We develop *SyncID-Pipe*, which seamlessly transfers the superiority of IFS to VFS, effectively boosting the video face swapping. 2) We propose *OmniFace*, the first video face swap-

ping framework based on DiT. 3) We introduce *IDBench-V*, a comprehensive benchmark tailored for the video face swapping task.

**Significance.** 1) *OmniFace* demonstrates superior generation performance compared to state-of-the-art methods. 2) We present a comprehensive study of the VFS task—including data, model, and benchmark. 3) Our proposed framework shows remarkable versatility and can be flexibly adapted to various swap-related tasks.

## 2 RELATED WORK

**Video Foundation Model.** The development of diffusion models Ho et al. (2020) has significantly advanced video foundation model research. Early latent diffusion methods Ho et al. (2022); Blattmann et al. (2023); Guo et al. (2023) extended Text-to-Image models with U-Net architectures to the video domain by incorporating temporal modules such as 3D convolutions and temporal attention. Rencently, the emergence of Diffusion Transformer(DiT) Peebles & Xie (2023b)-based methods for video generation has exhibited superior performance in quality and consistency. These methods Liu et al. (2024); Yang et al. (2024); Wan et al. (2025); Kong et al. (2024); Ma et al. (2025) employ powerful scaling transformers to generate longer and higher-quality videos. In addition to common Text-to-Video and Image-to-Video models, a growing number of keyframe interpolation models Gao et al. (2025) have emerged (e.g., First-Last-Frame models), paving the way for various downstream video generation tasks.

**Face Swapping.** Early image face swapping primarily focused on GAN-based models, such as FSGAN Nirkin et al. (2019; 2022), FaceShifter Li et al. (2020), HifiFace Wang et al. (2021), and SimSwap Chen et al. (2020). More recently, with the rapid development of diffusion models, several diffusion-based image face swapping models have emerged, including DiffFace Kim et al. (2022), DiffSwap Zhao et al. (2023), FaceAdapter Han et al. (2024), ReFace Baliah et al. (2024), and DreamID Ye et al. (2025), all of which have achieved promising results. Compared to image face swapping, video face swapping is still in its nascent stages. VividFace Shao et al. (2024) models video face swapping as a conditional inpainting task and proposes the first diffusion-based framework. DynamicFace Wang et al. (2025) incorporates precise and disentangled facial conditions for flexible and accurate control. HiFiVFS Wang et al. (2021) introduces an additional attribute extraction module to capture fine-grained attribute features. CanonSwap Luo et al. (2025) proposes a canonical space, performing face swapping within this space before projecting back to the real domain. While these works have improved the temporal consistency and quality of generated results, their performance in terms of identity similarity and attribute preservation remains largely unsatisfactory due to a lack of explicit supervision. In contrast, we significantly advance video face swapping by leveraging the superiority of image face swapping to construct explicit supervision.

## 3 METHODOLOGY

We propose a comprehensive framework to boost Video Face Swapping (VFS) by harnessing the prowess of Image Face Swapping (IFS). We first introduce a novel data curation pipeline *SyncID-Pipe*, which constructs bidirectional ID quadruplets to bridge the gap between VFS and IFS (Sec. 3.1). Building upon paired data, we develop *OmniFace*, the first Diffusion Transformer (DiT)-based video face swapping framework employing a core Modality-Aware Conditioning module to discriminatively inject conditions from multiple modalities (Sec. 3.2). To further enhance visual realism and identity consistency under challenging scenarios, we design a Synthetic-to-Real Curriculum mechanism and an Identity-Coherence Reinforcement Learning strategy during training (Sec. 3.3). Moreover, our proposed framework exhibits exceptional versatility (Sec. 3.4).

### 3.1 SYNCID-PIPE

IFS has demonstrated better performance in identity and attribute preservation compared to VFS. The fundamental difference between IFS and VFS lies in the dynamic nature of video, which requires consistent preservation of motion and expression across frames. This observation inspires us to explore bridging the gap between image and video domains by supplementing dynamic signals, thereby leveraging the strengths of IFS to significantly boost VFS performance.

#### 3.1.1 IDENTITY-ANCHORED VIDEO SYNTHESIZER

Building on this insight, we introduce a simple yet effective Identity-Anchored Video Synthesizer (IVS) to generate pair data for explicitly supervised training. As shown in Fig. 2, the IVS is trained to reconstruct a portrait video $V_r$ by leveraging its extracted pose sequence $p$. This is achieved by

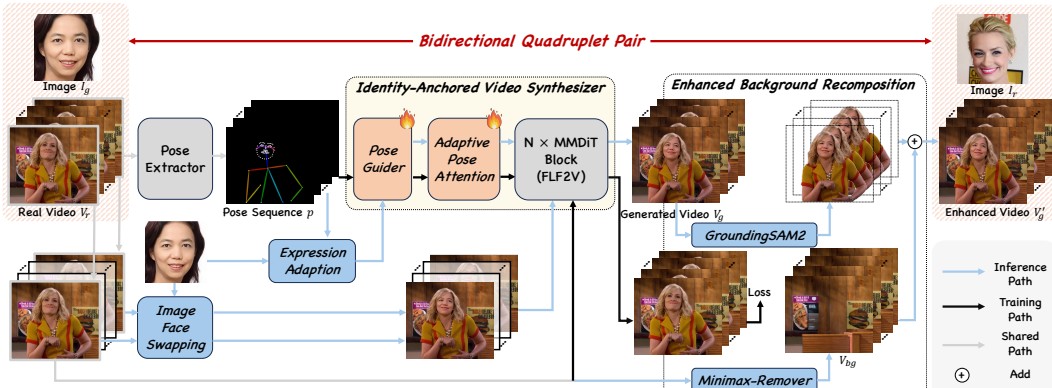

Figure 2: **Overview of SyncID-Pipe.** We pre-train the Identity-Anchored Video Synthesizer and combine it with the Image Face Swapping model to construct Bidirectional Quadruplet Pair data.

conditioning a First-Last-Frame video foundation model (FLF2V) Gao et al. (2025) on the initial and final frames of $V_r$, along with its pose sequence $p$. Such reconstruction-based training enables large-scale video pre-training. To minimize modifications to the foundation model and facilitate seamless integration of its pre-trained motion priors with the face swapping framework (detailed in Sec. 3.2), we introduce an Adaptive Pose-Attention mechanism to inject motion information.

**Adaptive Pose-Attention.** We employ a lightweight Pose Guider composed of several simple convolutional layers to extract pose features and align them with the dimension of the latent feature. To ensure precise spatiotemporal alignment between the pose sequence and the noisy latent video, we reuse the Rotary Position Embedding (RoPE) Su et al. (2024) indices from the noisy latents for the pose condition, which can help maintain the alignment. In each DiT block, we introduce two trainable linear layers $\mathbf{W}'_k$ and $\mathbf{W}'_v$ to align pose features $\mathbf{P}$ with latent feature $\mathbf{Z}$. Formally, the output of the Pose-Attention $\mathbf{Z}_{new}$ is:

$$\mathbf{Z}_{\text{new}} = \text{Softmax}\left(\frac{\mathbf{Q}\mathbf{K}^\top}{\sqrt{d}}\right)\mathbf{V} + \lambda \cdot \text{Softmax}\left(\frac{\mathbf{Q}(\mathbf{K}')^\top}{\sqrt{d}}\right)\mathbf{V}', \tag{1}$$

where $\mathbf{Q} = \mathbf{Z}\mathbf{W}_q$, $\mathbf{K} = \mathbf{Z}\mathbf{W}_k$, and $\mathbf{V} = \mathbf{Z}\mathbf{W}_v$ are derived from the frozen DiT layers, while $\mathbf{K}' = \mathbf{P}\mathbf{W}'_k$ and $\mathbf{V}' = \mathbf{P}\mathbf{W}'_v$ are from our trainable pose-adapter layers. The hyperparameter $\lambda$ controls the strength and flexibility of incorporating pose features. We train IVS by collecting a large-scale portrait dataset and extracting pose sequences. The model is optimized using Flow Matching Lipman et al. (2023). Through the aforementioned training, IVS can generate a video consistent with the content of the given keyframes and the actions specified by the pose sequence. This video is subsequently utilized for constructing bidirectional ID quadruplets.

### 3.1.2 BIDIRECTIONAL ID QUADRUPLETS CONSTRUCTION

Leveraging the identity preservation and dynamic attribute controllability of our designed IVS model, we effectively bridge the gap between IFS and VFS. Building upon this capability, we construct bidirectional ID quadruplet training data, anchored by IFS, to enable explicit supervision. As illustrated in Fig. 2, for a source image-video pair $(I_r, V_r)$ with identity ID A and a target image $(I_g)$ with identity ID B, we first utilize a state-of-the-art IFS model Ye et al. (2025) to transfer ID B onto the first and last frames of $V_r$. This process yields high-quality reference frames ($I_{ref1}$, $I_{ref2}$). Subsequently, these reference frames, along with the retargeted pose sequence, are fed into the pre-trained IVS module to synthesize $V_g$, a video of ID B. The resulting bidirectional ID quadruplet is formatted as $\{I_r, V_r, I_g, V_g\}$, where $\{I_r, V_g, V_r\}$ constitutes the forward-generated paired data, and $\{I_r, V_r, V_g\}$ represents the backward-real paired data. To further enhance the diversity and practicality of the training data, we implement the following strategies:

**Source Data Curation.** To ensure the robustness of *OmniFace* across diverse and challenging scenarios, we carefully curate source videos encompassing varied makeup styles, extreme lighting conditions, and other adversarial settings. Furthermore, we incorporate talking-head datasets to enhance the ability of the model to preserve subtle facial expressions and accurate lip-synchronization.

**Expression Adaptation.** Considering that simply using the pose sequence of the source video to drive the generation of the target video leads to identity-expression entanglement and thus causes identity leakage, we use an expression adaptation module to decouple identity and expression, allowing for high-quality expression transfer. During inference, a 3D face reconstruction model Wang

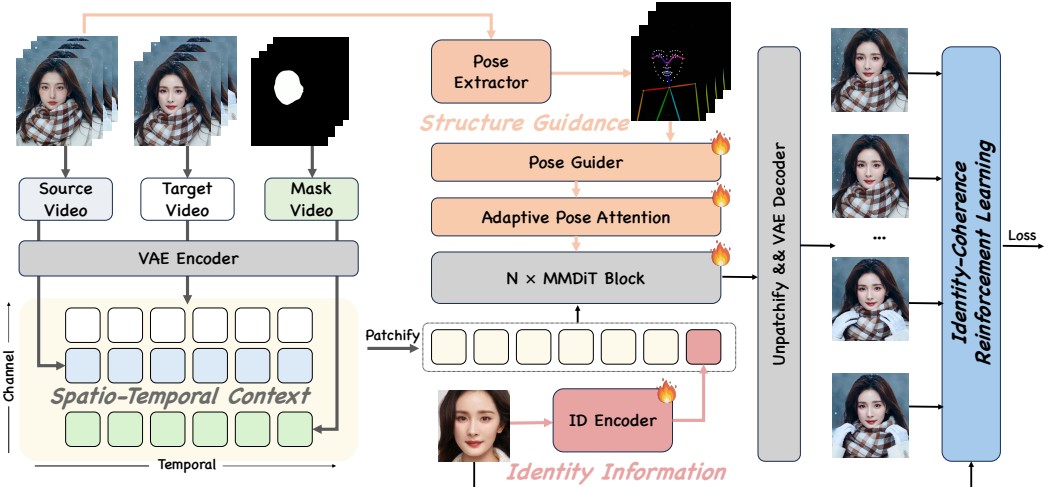

Figure 3: **Overview of *OmniFace* framework.** We design customized injection mechanisms for Spatio-Temporal Context, Structural Guidance, and Identity Information, respectively.

et al. (2024) extracts identity coefficient from $I_g$, and the expression and pose coefficients from each $V_r$ frame. We then recombine identity of $I_g$ with expression and pose of each $V_r$ frame to reconstruct a new 3D face model. Projecting this model yields retargeted facial landmarks, which replace the original ones in the pose sequence for final inference.

**Enhanced Background Recomposition.** Static keyframe-driven IVS often produces videos with background inconsistencies relative to the source videos, particularly with significant background motion. To enhance the applicability of the model in real-world dynamic scenarios, we design an Enhanced Background Recomposition module. As shown in Fig. 2, for forward-generated paired data $\{I_r, V_g, V_r\}$, we first extract foreground masks from $V_r$ and $V_g$ using SAM2 Ravi et al. (2024). We then utilize MinimaxRemover Ravi et al. (2024) to remove the foreground from $V_r$, yielding a clean background video $V_{bg}$. Subsequently, the foreground from $V_g$ is pasted back into $V_{bg}$ to form enhanced video $V_g'$. A feathering operation is then applied at the foreground edges to achieve a smooth and natural blending. Crucially, in the final paired data obtained this way, the supervision video remains the real video $V_r$. This prevents the model from learning artifacts introduced during the background pasting process. By specifically augmenting these data, we aim to improve the capabilities of the model in background preservation.

## 3.2 OMNIFACE FRAMEWORK

VFS aims to seamlessly transfer identity information from a source face to a target video, while preserving attributes of the target video, such as pose, expression, and background. A core challenge in designing such a model involves effectively injecting and simultaneously disentangling and fusing different types of information, including identity and attributes. To address this, we develop ***OmniFace***, the first video face swapping framework based on Diffusion Transformer (DiT) models, as illustrated in Fig. 3. Central to this framework is the Modality-Aware Conditioning (MC) mechanism, designed to efficiently inject multiple conditions, tailored to their respective needs.

### 3.2.1 MODALITY-AWARE CONDITIONING.

To make conditions disentangled from each other, the MC mechanism decomposes the conditions for video face swapping into three distinct types. The details are as follows:

**Spatio-Temporal Context Module.** To provide the contextual reference information to be retained (e.g., background, lighting), we inject both the reference video and the dilated face mask into the model. Since these conditions must align precisely with the latent noise in both spatial and temporal dimensions, we concatenate them with latent of source video along the channel dimension.

**Structural Guidance Module.** To achieve fine-grained preservation of dynamic attributes, we incorporate the pose condition as structural guidance. To this end, we employ the Pose-Attention mechanism and initialize corresponding parameters with a pre-trained model described in Sec. 3.1.1, which fully leverages the prior from the IVS model. The strategy enables efficient structural control while avoiding disruption to the high-level features being processed by the DiT.

**Identity Information Module.** In contrast to the context and structural information, identity condition represents high-level semantic features and requires comprehensive spatiotemporal interaction. We first employ a dedicated ID encoder to encode the reference identity into ID embeddings. And then we concatenate them with the latent noise after patchification along the token dimension, which enables full interaction with latent features through the DiT's inherent attention mechanism.

In summary, the spatio-temporal context module provides required attribute information and leverages the mask to indicate facial regions to the model. The structural guidance module further enhances motion attributes, improving the preservation of exaggerated movements and fine-grained expressions. Finally, the identity information module efficiently injects identity-related features.

### 3.3 OMNIFACE TRAINING PIPELINE

A significant challenge in training video face swapping models involves balancing identity similarity, attribute preservation, visual realism, and ensuring temporal consistency of identity. To mitigate the issue, we introduce a Sync-to-Real Curriculum mechanism along with an Identity-Coherence Reinforcement Learning strategy. The training process is divided into the following stages:

**Synthetic Training:** In our constructed bidirectional ID quadruplet data $\{I_r, V_r, I_g, V_g\}$, $\{I_r, V_g, V_r\}$ represents the forward-generated paired data, while $\{I_g, V_r, V_g\}$ represents the backward-real paired data. In this initial stage, we train our model using the forward-generated paired data. Since $V_g$ is synthesized by the IVS module, it remains distributionally consistent with our underlying video foundation model. (as further elaborated in the supplementary material). This alignment significantly accelerates model convergence and enables the attainment of higher identity similarity, yielding superior similarity compared to direct training with backward-real paired data.

**Real Augmentation Training:** The previous stage yields a VFS model with high identity similarity, however, its realism and background preservation remain limited due to the fact that the forward-generated paired data is model-generated and inherently lacks fidelity in background and realism. Therefore, we introduce a real augmentation training stage to fine-tune the model, which uses backward-real paired data $\{I_g, V_r, V_g'\}$ augmented by the Enhanced Background Recomposition strategy introduced in Sec. 3.1.2. Experiments show that this stage enables the model to maintain strong identity similarity while achieving excellent background.

**Identity-Coherence Reinforcement Learning.** While the aforementioned mechanisms establish a robust baseline, temporal identity consistency remains a challenge in complex scenarios, particularly in videos with significant motions. We observe that identity similarity fluctuates, remaining high in frontal views and mild movements but degrading considerably during profile views or intense actions. To specifically address this final-mile problem, as shown in Fig. 3, we introduce the Identity-Coherence Reinforcement Learning (IRL) mechanism, inspired by Ding et al. (2024); Liu et al. (2025). The core insight behind IRL is to incentivize the model to focus its learning capacity on difficult frames. We formalize this as a policy optimization problem, where the model's generative process is treated as a policy, $\pi_\theta$, that aims to produce video frames with maximal identity fidelity. Consequently, frames with low identity fidelity are inherently more valuable as learning signals, as they represent the greatest potential for policy improvement. Instead of learning a complex Q-function via temporal difference updates in some methods Schulman et al. (2017); Lillicrap et al. (2015), we leverage the specific nature of the video face-swapping task to define an efficient and explicit Q-value. Given a state $y$ representing the conditional inputs and an action $\hat{x}_0$ corresponding to the generated video frame, we define the Q-value as:

$$Q(y, \hat{x}_0) = \frac{1}{\cos(E(\hat{x}_0), E(I_t)) + \delta}, \tag{2}$$

where $E(\cdot)$ denotes the feature embedding extracted by Deng et al. (2019a), $\cos(\cdot, \cdot)$ is the cosine similarity, $I_t$ is the target identity image, and $\delta$ is a small constant for numerical stability.

As implemented in our pipeline, we first perform a full sampling pass without backpropagation to generate a video and compute the Q-value for each frame. These frame-wise Q-values are then averaged to obtain $Q_c$ for each VAE-encoded chunk. These $Q_c$ are then used as weights for the flow matching loss Lipman et al. (2023) during the IRL training step:

$$\mathcal{L}_{\text{IRL}}(\theta) = \sum_{c=1}^{C} \mathbb{E}_{t,\epsilon} \left[ Q_c \cdot \|(\mathbf{z}_c - \epsilon) - \mathbf{v}_\theta \left((1-t)\mathbf{z}_c + t\epsilon, t, y\right)\|^2 \right], \tag{3}$$

where the loss is summed over all $C$ VAE-encoded chunks in the video. This loss can also be interpreted as a reward-weighted likelihood maximization objective as formally proven in Sec. A.5.

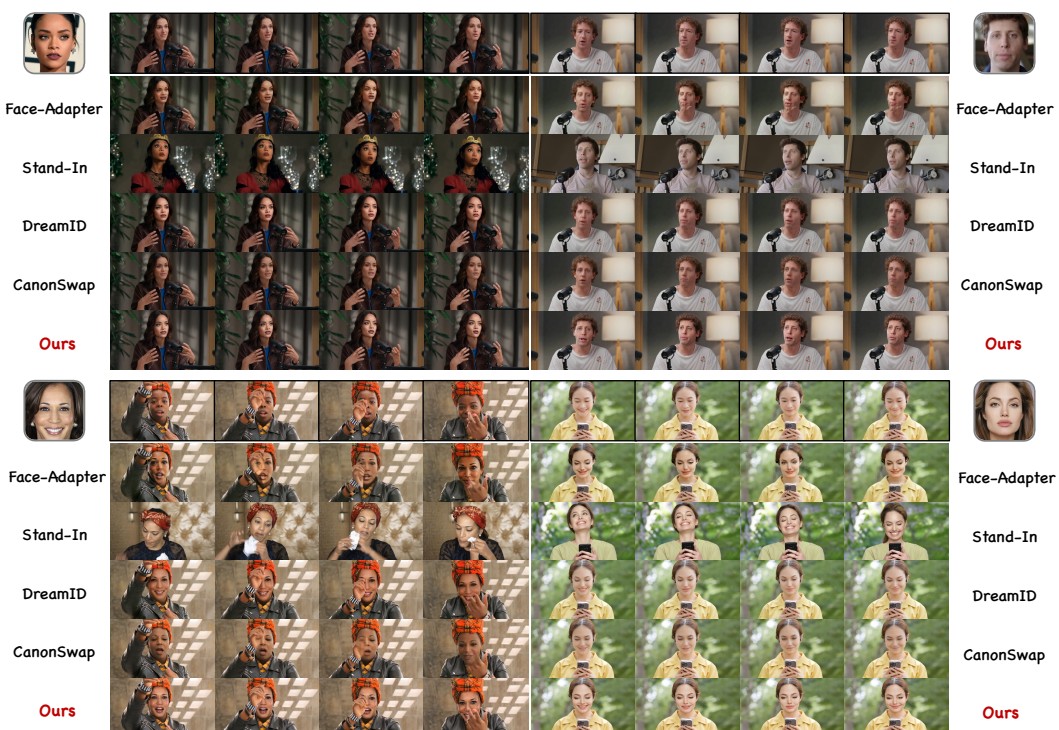

Figure 4: **Qualitative comparisons** with state-of-the-art methods. Please zoom in for more details.

Table 1: **Quantitative comparisons** with baseline methods.

| Method | Identity Consistency | | | | Attribute Preservation | | | | Video Quality | |
|---|---|---|---|---|---|---|---|---|---|---|
| | ID-Arc ↑ | ID-Ins ↑ | ID-Cur ↑ | Variance ↓ | Pose ↓ | Expression ↓ | Background ↑ | Subject ↑ | FVD ↓ | Smoothness ↑ |
| FSGAN Nirkin et al. (2022) | 0.435 | 0.466 | 0.441 | 0.0069 | 7.415 | 3.463 | 0.904 | 0.919 | 6.582 | 0.982 |
| REFace Baliah et al. (2024) | 0.472 | 0.471 | 0.474 | 0.0191 | 5.102 | 2.785 | 0.909 | 0.913 | 7.084 | 0.988 |
| Face-Adapter Han et al. (2024) | 0.440 | 0.496 | 0.450 | 0.0081 | 5.156 | 3.037 | 0.942 | 0.945 | 3.460 | 0.988 |
| DreamID Ye et al. (2025) | 0.616 | 0.702 | 0.664 | 0.0058 | 3.013 | 2.930 | 0.940 | 0.951 | 3.108 | 0.989 |
| Stand-In Xue et al. (2025) | 0.403 | 0.403 | 0.367 | 0.0057 | 19.819 | 2.995 | 0.931 | 0.951 | 3.368 | 0.982 |
| CanonSwap Luo et al. (2025) | 0.397 | 0.431 | 0.407 | 0.0030 | **2.430** | 2.477 | 0.950 | 0.954 | **2.176** | 0.991 |
| **Ours** | **0.659** | **0.713** | **0.688** | **0.0029** | 2.446 | **2.430** | **0.951** | **0.966** | 2.243 | **0.992** |

## 3.4 OMNIFACE VERSATILITY

Notably, our proposed *OmniFace* extends beyond face swapping task. By constructing explicit paired data for various human-centric swapping tasks—such as outfit, accessory, and hairstyle swapping—simply through replacing the Image Face Swapping (IFS) model with a general-purpose image editing model (e.g., Nano banana Banana (2025)) within the SyncID-Pipe, *OmniFace* can be extended to a wider range of swap category tasks. We will detail this extensibility in Sec. 4.5.

## 4 EXPERIMENTS

### 4.1 SETUP

**IDBench-V.** We introduce *IDBench-V*, a new comprehensive benchmark for video face swapping. The benchmark comprises 200 real-world source video-target image pairs, covering a diverse range of challenging scenarios. These include small faces, extreme head poses, severe occlusions, complex and dynamic expressions, and cluttered multi-person scenes. IDench-V provides a rigorous and holistic platform for evaluation in real-world usage scenarios.

**Implementation Details.** We choose OpenHumanVid Li et al. (2024) as the training set and subsequently filter it based on ID similarity to create paired videos of the same identity. Training our IVS module requires 3,000 GPU-hours, while the entire *OmniFace* training consumes approximately 6,000 GPU-hours, both on NVIDIA A100 GPUs. Refer to Sec. A.3.2 for more details.

**Baselines.** We conduct comparisons against existing face swapping state-of-the-art (SOTA) models on *IDBench-V*. For image face swapping, we compare with FSGAN Nirkin et al. (2022), REFace Baliah et al. (2024), Face-Adapter Han et al. (2024), and DreamID Ye et al. (2025), noting that these image-based methods achieve video face swapping by processing frames individually.

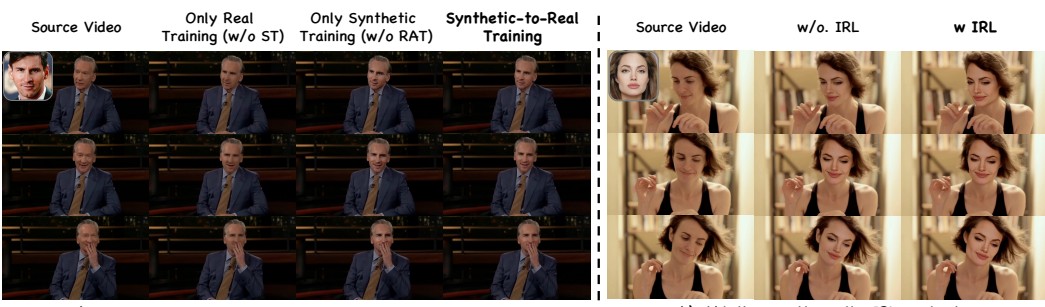

Figure 5: **Ablation studies** of *OmniFace*.

For video face swapping, we evaluate against Stand-In Xue et al. (2025) and CanonSwap Luo et al. (2025). Due to the unavailability of open-source code for VividFace Shao et al. (2024) and DynamicFace Wang et al. (2025), we perform a qualitative comparison using videos from their respective demos in Sec. A.7.

**Evaluation Metrics** We evaluate the performance of various video face swapping methods across three key dimensions: Identity Consistency, Attribute Preservation, and Video Quality. For Identity Consistency, we employ ArcFace Deng et al. (2019a), InsightFace The InsightFace Team, and CurricularFace Huang et al. (2020) to compute ID similarity. To quantify temporal stability, we additionally calculate the variance of these frame-wise similarities. Attribute Preservation is assessed by evaluating the fidelity of pose and expression transferred from the driving video.(details are provided in the supplementary material) Furthermore, we incorporate three metrics from VBench Huang et al. (2024): background consistency, subject consistency, and motion smoothness, which respectively evaluate the consistency of the background, the primary subject, and the overall motion. Finally, for Video Quality, we evaluate perceptual video quality in unpaired scenarios using the Fréchet Video Distance (FVD) Ge et al. (2024) with a ResNext Xie et al. (2016) feature extractor.

## 4.2 QUANTITATIVE COMPARISONS

**Metric Evaluation.** As shown in Tab. 1, *OmniFace* comprehensively outperforms state-of-the-art models in terms of identity similarity metrics. Regarding attribute preservation, *OmniFace* is optimal across almost all metrics, with only a slight inferiority to CanonSwap in terms of pose. It is worth noting that CanonSwap, due to its very low identity similarity, results in minimal alteration to the original video, thereby exhibiting good attribute preservation and video quality. Our proximity to CanonSwap in attribute preservation demonstrates our model's excellent capability in this regard, while its identity similarity is significantly higher than CanonSwap. This superiority is attributed to our ID quadruplet effectively transferring the high identity similarity from DreamID to VFS. Interestingly, owing to the effectiveness of our training strategy, our identity similarity even slightly surpasses that of DreamID. In terms of video quality, our model also achieves outstanding results, showing substantial improvements compared to IFS models such as REFace, Face-Adapter, and DreamID. This collectively demonstrates that our model not only achieves high identity similarity and robust attribute preservation but also generates high-quality videos.

**User Study.** We invited 19 volunteers to conduct a human evaluation of the models on ID-Bench. Each sample was rated across three dimensions: Identity Similarity, Attribute Preservation, and Video Quality, with scores ranging from 1-5. As presented in Tab. 2, show that our model achieved the best performance across all metrics, thereby demonstrating the superior capabilities of our model.

Table 2: **User study** of *OmniFace*.

| Method | ID Sim ↑ | Attr ↑ | Quality ↑ |
|---|---|---|---|
| REFace Baliah et al. (2024) | 1.45 | 2.15 | 1.11 |
| Face-Adapter Han et al. (2024) | 2.17 | 2.93 | 1.14 |
| DreamID Ye et al. (2025) | 3.78 | 3.89 | 3.06 |
| Stand-In Xue et al. (2025) | 2.45 | 1.60 | 2.91 |
| Canonswap Luo et al. (2025) | 1.99 | 3.91 | 3.42 |
| **Ours** | **3.85** | **4.22** | **4.15** |

## 4.3 QUALITATIVE ANALYSIS

We conduct a qualitative comparison with VFS methods CanonSwap and Stand-In, as well as IFS methods Face-Adapter and DreamID. As illustrated in Fig. 4, *OmniFace* demonstrates excellent performance across identity similarity, expression preservation, background preservation, and occlusion. In the first two cases presented in the first row, our method demonstrates superiority in

Figure 6: **Versatility** of our methods. Please zoom in for more details.

identity similarity compared to other approaches, consistently across both male and female subjects. Specifically, our identity similarity is significantly better than that of Face-Adapter, Stand-In, and CanonSwap. Perceptually, the identity similarity is close to DreamID, however, we observe that our IVS module, by incorporating dynamic expression information, leads to superior expression performance compared to DreamID. Stand-In, which utilizes an inpainting approach for face swapping, introduces substantial alterations to the original video. In the second row, the left case highlights the robust performance of our model under occlusion, outperforming all other models. The right case showcases the superiority of our model in handling complex expressions.

### 4.4 ABLATION STUDIES

To demonstrate the effectiveness of our proposed method, we conduct the following ablation studies: a). Directly training using the self-reconstruction inpainting-based approach (w/o Quadruplet); b). Training exclusively with backward-real paired data (w/o ST); c). Training

Table 3: **Ablation study** of *OmniFace*.

| Method | ID-Arc ↑ | Variance ↓ | Pose ↓ | Expression ↓ | FVD ↓ |
|---|---|---|---|---|---|
| a) w/o Quadruplet | 0.510 | 0.0036 | 2.468 | 2.432 | 2.242 |
| b) w/o ST | 0.604 | 0.0035 | 2.742 | 2.445 | **2.145** |
| c) w/o RAT | 0.657 | 0.0042 | 2.557 | 2.443 | 3.845 |
| d) w/o IRL | 0.631 | 0.0041 | 2.687 | 2.488 | 2.206 |
| **e) Ours** | **0.659** | **0.0029** | **2.446** | **2.430** | 2.243 |

solely with forward-generated paired data (w/o RAT); d). Training without the Identity-Coherence Reinforcement Learning stage (w/o IRL). As shown in Tab. 3 a), Traditional inpainting-based method yields significantly lower identity similarity, demonstrating the effectiveness of *SyncID-Pipe* to construct explicitly supervised data to bridge the gap between VFS and IFS. b). As demonstrated in Fig. 5 a), the w/o ST setting yields higher realism but lower identity similarity (i.e., a good FVD score but poor ID-Arc). Conversely, w/o RAT achieves superior identity similarity but at the cost of realism (i.e., a good ID-Arc score but poor FVD). In contrast, our Sync-to-real training strategy(line w/o IRL) ultimately strikes a good balance, maintaining high identity similarity while preserving realism. Furthermore, the IRL mechanism significantly enhances identity similarity under complex motions, leading to a noticeable improvement. A comparison between lines (d) and (e) reveals that IRL not only improves ID-Arc to some extent but also substantially reduces variance, which represents the consistency of inter-frame similarity. As depicted in Fig. 5 b), the top and bottom frames present profile views, while the middle frame shows a frontal view. Without IRL, the model performs well on frontal views but poorly on profile views. However, IRL substantially boosts identity similarity in profile views, as exemplified by the topmost frame.

### 4.5 VERSATILITY

As shown in Fig. 6, by expanding training data, our model can be extended to a wider range of human-centric swapping tasks, including accessory, outfit, headphone, and hairstyle swapping.

## 5 CONCLUSION

This work presents a comprehensive framework for Video Face Swapping (VFS). Our *SyncID-Pipe* data pipeline effectively transfers the superiority of image face swapping to video, enabling *OmniFace*—the first DiT-based model for VFS—to achieve superior performance on our proposed comprehensive benchmark *IDBench-V*. The method demonstrates strong versatility and provides a systematic solution for high-fidelity VFS.

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

# A  APPENDIX

In the supplementary material, the sections are organized as follows:

- we provide LLM Usage statement in Sec. A.1.
- We provide the details of Flow Matching in Sec. A.2.
- We provide more details regarding parameters, datasets, inference, evaluation metrics and user study in Sec. A.3.
- We provide the details of our proposed benchmark *IDBench-V* in Sec A.4
- We provide theoretical justification for IRL mechanism in Sec. A.5.
- We provide the distribution visualization of synthetic data and real data in Sec. A.6.
- We provide more comparisons with baselines, more qualitative results in Sec. A.7.
- We provide limitations in Sec. A.8.
- We provide Ethical Considerations in Sec. A.9

## A.1  LLM USAGE STATEMENT

The core scientific ideas, methodology, experimental results, and conclusions presented in this paper are entirely the product of human authorship. A large language model was utilized solely as a language refinement tool, specifically to enhance the conciseness and clarity of the English text and to correct grammatical errors.

## A.2  PRELIMINARY

The Diffusion Transformer (DiT) Peebles & Xie (2023b) model employs a transformer as the denoising network to refine the diffusion latent. Our method inherits the video diffusion transformers trained using Flow Matching Lipman et al. (2023), which conducts the forward process by linearly interpolating between noise and data in a straight line. At the time step $t$, latent $\mathbf{z}_t$ is defined as: $\mathbf{z}_t = (1-t)\mathbf{z}_0 + t\epsilon$, where $\mathbf{z}_0$ is the clean video, and $\epsilon \sim \mathcal{N}(0,1)$ is the Gaussian noise. The model is trained to directly regress the target velocity:

$$\mathcal{L}_{\text{FM}} = \mathbb{E}_{t,\mathbf{z}_0,\epsilon}\left[\left\|(\mathbf{z}_0 - \epsilon) - \mathbf{v}_\theta(\mathbf{z}_t, t, y)\right\|^2\right], \tag{4}$$

where $\mathbf{v}_\theta$ refers to the diffusion model output and $y$ denotes condition.

## A.3  IMPLEMENTATION DETAILS

### A.3.1  INFERENCE DETAILS

To fully leverage the model's enhanced capabilities and maximize identity fidelity, strong Classifier-Free Guidance (CFG) Ho & Salimans (2022) is required. In the context of video face swapping, it is natural to leverage the guidance vector to steer the generation towards a high degree of identity similarity with the target. This guidance vector, $\mathbf{d}$, is defined as the difference between the velocity predictions with and without the target identity condition, $\mathcal{C}_{\text{id}}$:

$$\mathbf{d} = \mathbf{v}_\theta(z_t, \mathcal{C}_{\text{pose}}, \mathcal{C}_{\text{ref}}, \mathcal{C}_{\text{id}}) - \mathbf{v}_\theta(z_t, \mathcal{C}_{\text{pose}}, \mathcal{C}_{\text{ref}}, \emptyset), \tag{5}$$

where $\mathcal{C}_{\text{pose}}$ is the pose sequence, $\mathcal{C}_{\text{ref}}$ is the concatenation of the source video and its mask, and $\emptyset$ denotes the null identity condition. However, we observe that naively applying conventional CFG often introduces a pernicious trade-off: while identity similarity increases, it frequently leads to oversaturation and unrealistic artifacts. Motivated by prior work on guidance modification Sadat et al. (2024); Zhang et al. (2025), we propose ID Guidance Purification (IDGP). Our method decomposes the guidance vector $\mathbf{d}$ into parallel and orthogonal components relative to the normalized conditional prediction $\hat{\mathbf{v}}_{\text{cond}}$:

$$\mathbf{d}_\parallel = \langle \mathbf{d}, \hat{\mathbf{v}}_{\text{cond}} \rangle \hat{\mathbf{v}}_{\text{cond}}, \tag{6}$$

$$\mathbf{d}_\perp = \mathbf{d} - \mathbf{d}_\parallel, \tag{7}$$

where $\mathbf{v}_{\text{cond}} = \mathbf{v}_\theta(z_t, \mathcal{C}_{\text{pose}}, \mathcal{C}_{\text{ref}}, \mathcal{C}_{\text{id}})$ and $\hat{\mathbf{v}}_{\text{cond}} = \mathbf{v}_{\text{cond}}/\|\mathbf{v}_{\text{cond}}\|$. We empirically find that the orthogonal component, $\mathbf{d}_\perp$, is the primary contributor to the aforementioned artifacts. IDGP, therefore, adjusts the composition of the guidance by differentially re-weighting these components to create a purified guidance vector, $\mathbf{d}_{\text{IDGP}}$:

$$\mathbf{d}_{\text{IDGP}} = \alpha \cdot \mathbf{d}_\parallel + \frac{1}{\alpha} \cdot \mathbf{d}_\perp, \tag{8}$$

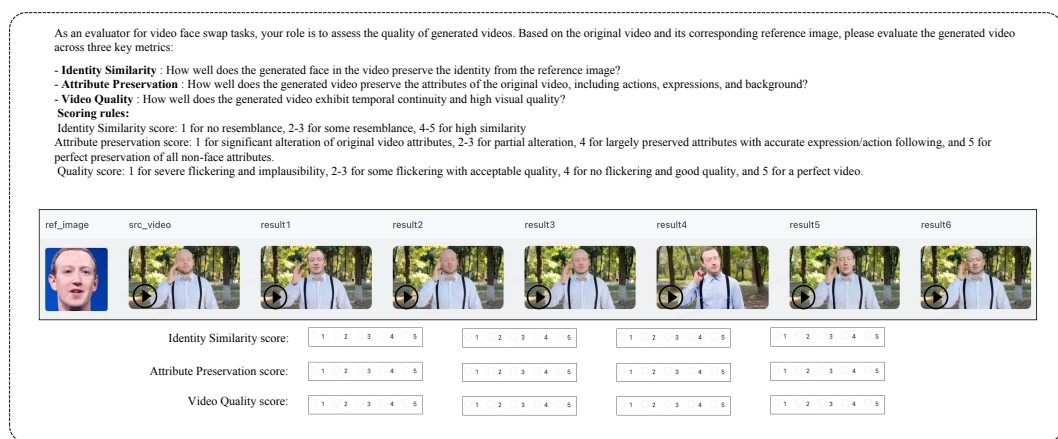

Figure 7: Demo of **User study**.

where $\alpha > 1$ is a hyperparameter that simultaneously amplifies the identity-preserving parallel signal and suppresses the artifact-inducing orthogonal signal. Finally, the purified guidance $\mathbf{d}_{\text{IDGP}}$ is applied to the conditional prediction with an overall guidance scale $s$:

$$\mathbf{v}_{\text{output}} = \mathbf{v}_{\text{cond}} + s \cdot \mathbf{d}_{\text{IDGP}}, \tag{9}$$

This process effectively purifies the guidance signal, enabling strong identity preservation without sacrificing realism.

### A.3.2 DETAILED PARAMETERS

We use the AdamW optimizer with a constant learning rate of $1.0 \times 10^{-5}$ for all training stages. The IVS module is trained on 1000 hours of video data. The Synthetic Training stage utilizes 100 hours of IVS-generated video as GT, followed by the Real Augmentation Training stage with 150 hours of real and synthetic data. Finally, the IRL stage is conducted on 10 hours of data selected for high ID variance. For inference, we use the Euler sampler with 8 steps, set $\lambda$ to 0.5 for IVS module, and set the guidance scale $s$ to 4.5, the parameter $\alpha$ in IDGP to 3 for *OmniFace*.

The training process for *OmniFace* commences with 50k iterations with a global batch size of 16 on exclusively synthetic ground truth (GT) data $V_g$ to rapidly establish a strong baseline for ID similarity. Subsequently, the model is fine-tuned for an additional 80k iterations with a global batch size of 32 on a hybrid dataset of real GT $V_r$ and synthetic GT $V_g$ to enhance photorealism. The training regimen culminates in the application of our IRL, where the model is further refined on a curated subset of data from the previous stages—specifically, samples exhibiting high variance in ID similarity. The Synthetic Training stage requires 600 GPU-hours. The Real Augmentation Training stage takes 1400 GPU-hours. The final IRL stage, which involves online inference, consumes 4000 GPU-hours. The entire training for *OmniFace* requires approximately 6,000 GPU-hours on NVIDIA A100 GPUs.

### A.3.3 EVALUATION METRICS

We evaluate the performance of various video face swapping methods across three key dimensions: Identity Consistency, Attribute Preservation, and Video Quality. We employ ArcFace Deng et al. (2019a), InsightFace The InsightFace Team, and CurricularFace Huang et al. (2020) to extract face embeddings and compute the cosine similarity with the target identity image. Additionally, we calculate the variance of these frame-wise similarities to quantify temporal stability. Regarding Attribute Preservation, we assess the fidelity of pose and expression transferred from the driving video. This is achieved by computing the L2 distance between the generated frames and the driving frames in terms of head pose, estimated by HopeNet Ruiz et al. (2018), and expression coefficients, extracted via Deep3DFaceRecon Deng et al. (2019b).

### A.3.4 USER STUDY

Fig. 7 illustrates the interface of our user study. Each evaluator was presented with a source video, a reference image, and six anonymized videos generated by different models. Evaluators were instructed to rate each generated video across three dimensions—identity similarity, attribute preserva-

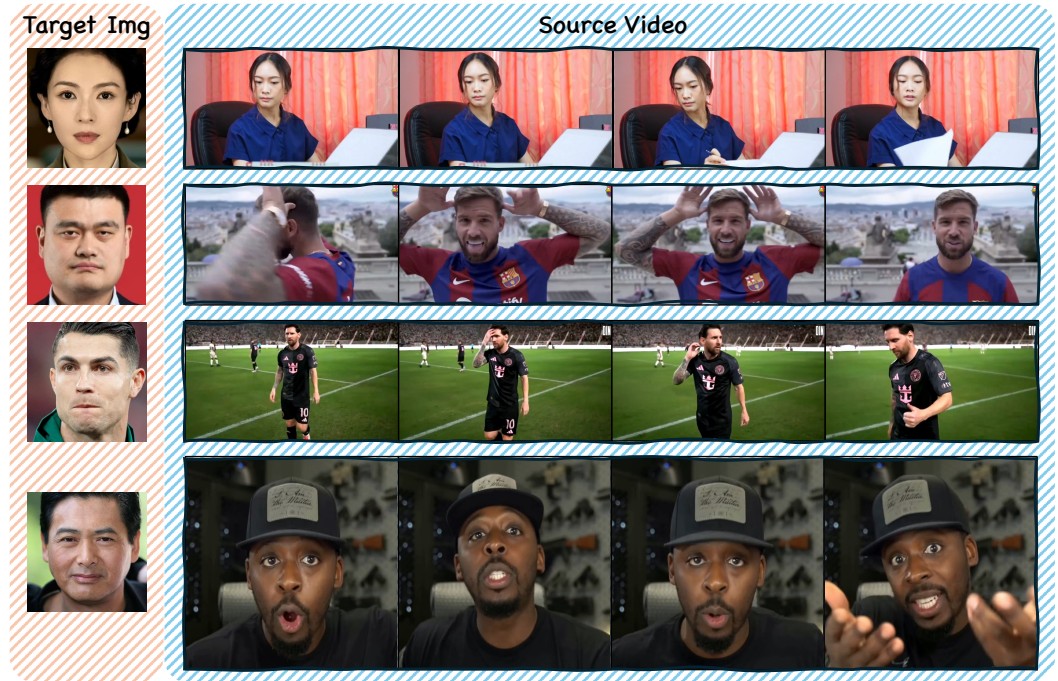

Figure 8: Examples of **IDBench-V**.

tion, and video quality—using a 1-to-5-point scale for each dimension. We recruited 19 evaluators, and their final scores were averaged to obtain the overall evaluation.

### A.4 IDBENCH-V DETAILS

To address the lack of a benchmark for Video Face Swapping (VFS), we introduce a comprehensive benchmark, **IDBench-V**, which consists of 200 videos paired with meticulously selected ID images. As shown in Fig. 8, our collected videos span diverse categories, including small faces, extreme head poses, severe occlusions, complex and dynamic expressions, and cluttered multi-person scenes.

### A.5 THEORETICAL JUSTIFICATION FOR IRL

**Theorem 1** (IRL as Reward-Weighted Likelihood Maximization). *Minimizing the IRL loss, $\mathcal{L}_{IRL}$, is equivalent to maximizing the reward-weighted log-likelihood objective, $J(\theta)$.*

*Proof.* Let $p_{\text{data}}(\mathbf{z}|y)$ be the true data distribution conditioned on $y$, and $\pi_\theta(\mathbf{z}|y)$ be the model distribution. The reward-weighted maximum likelihood objective is:

$$J(\theta) \triangleq \mathbb{E}_{\mathbf{z}\sim p_{\text{data}}}[Q(\mathbf{z}) \log \pi_\theta(\mathbf{z}|y)]$$

Our proposed loss is:

$$\mathcal{L}_{\text{IRL}}(\theta) \triangleq \mathbb{E}_{\mathbf{z}\sim p_{\text{data}}}[Q(\mathbf{z}) \cdot \mathcal{L}_{\text{FM}}(\mathbf{z};\theta)]$$

where $\mathcal{L}_{\text{FM}}(\mathbf{z};\theta) = \mathbb{E}_{t,\epsilon}[\|v_\theta((1-t)\mathbf{z} + t\epsilon, t, y) - (\epsilon - \mathbf{z})\|^2]$ is the standard flow matching loss for a sample $\mathbf{z}$. Our goal is to show $\nabla_\theta J(\theta) \propto -\nabla_\theta \mathcal{L}_{\text{IRL}}(\theta)$. Since $Q(\mathbf{z})$ is a pre-computed scalar for each sample, it acts as a constant weight inside the expectation. The proof thus simplifies to showing the equivalence for the unweighted objectives, i.e., $\nabla_\theta \mathbb{E}[\log \pi_\theta] \propto -\nabla_\theta \mathbb{E}[\mathcal{L}_{\text{FM}}]$. The KL divergence from $\pi_\theta$ to $p_{\text{data}}$ is $D_{\text{KL}}(p_{\text{data}}\|\pi_\theta) = \mathbb{E}_{p_{\text{data}}}[\log p_{\text{data}}] - \mathbb{E}_{p_{\text{data}}}[\log \pi_\theta]$. As the entropy of the data distribution is constant w.r.t. $\theta$, we have:

$$\nabla_\theta \mathbb{E}_{\mathbf{z}\sim p_{\text{data}}}[\log \pi_\theta(\mathbf{z}|y)] = -\nabla_\theta D_{\text{KL}}(p_{\text{data}}\|\pi_\theta)$$

From the theory of generative flow matching Lipman et al. (2023), minimizing $\mathcal{L}_{\text{FM}}$ is equivalent to minimizing $D_{\text{KL}}(p_{\text{data}}\|\pi_\theta)$, which implies their gradients are aligned:

$$\nabla_\theta \mathcal{L}_{\text{FM}}(\theta) \propto \nabla_\theta D_{\text{KL}}(p_{\text{data}}\|\pi_\theta)$$

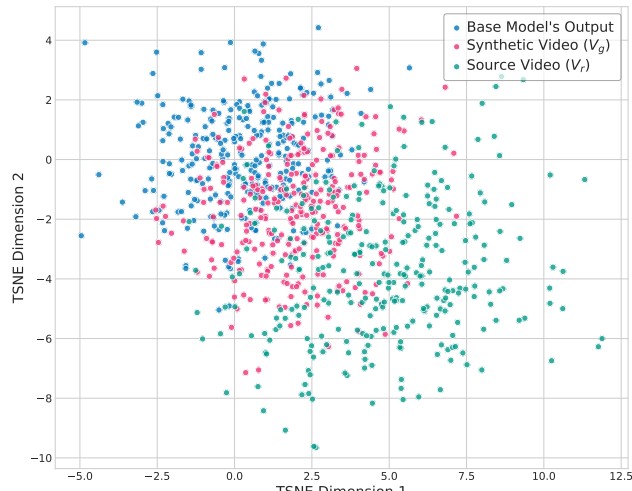

Figure 9: **T-SNE Visualization of Latent Space Distributions.**

Combining these two relations, we obtain the core proportionality:

$$\nabla_\theta \mathbb{E}_{\mathbf{z} \sim p_{\text{data}}}[\log \pi_\theta(\mathbf{z}|y)] \propto -\nabla_\theta \mathcal{L}_{\text{FM}}(\theta)$$

Extending this to the weighted case by applying the sample-wise weight $Q(\mathbf{z})$:

$$\begin{aligned}
\nabla_\theta J(\theta) &= \mathbb{E}_{\mathbf{z} \sim p_{\text{data}}}[Q(\mathbf{z})\nabla_\theta \log \pi_\theta(\mathbf{z}|y)] \\
&\propto \mathbb{E}_{\mathbf{z} \sim p_{\text{data}}}[Q(\mathbf{z}) \cdot (-\nabla_\theta \mathcal{L}_{\text{FM}}(\mathbf{z};\theta))] \\
&= -\nabla_\theta \mathbb{E}_{\mathbf{z} \sim p_{\text{data}}}[Q(\mathbf{z}) \cdot \mathcal{L}_{\text{FM}}(\mathbf{z};\theta)] \\
&= -\nabla_\theta \mathcal{L}_{\text{IRL}}(\theta)
\end{aligned}$$

This establishes that a gradient ascent step on $J(\theta)$ corresponds to a gradient descent step on $\mathcal{L}_{\text{IRL}}(\theta)$.
□

### A.6 T-SNE VISUALIZATION

To validate our claim that synthetic videos ($V_g$) are inherently distribution-aligned with our DiT-based architecture, we analyze their latent space representations. We sampled 300 videos respectively from three domains: real source videos ($V_r$), synthetic videos ($V_g$), and outputs from the base DiT model. Each video was encoded into a latent vector using a pre-trained VAE, and the resulting representations are visualized in 2D using t-SNE Maaten & Hinton (2008). As shown in Figure 9, the visualization reveals a clear structural relationship. The distribution of synthetic videos ($V_g$, pink) demonstrates a high degree of overlap with that of the base model's output (blue), forming a cohesive cluster. In contrast, the real source videos ($V_r$, green) occupy a more distinct and dispersed region of the latent space. This provides strong visual evidence that our IVS module generates data that is already well-aligned with the model's target distribution. This inherent alignment explains the accelerated convergence and improved performance observed when training with synthetic data, as it effectively narrows the domain gap from the outset.

### A.7 MORE VISUAL RESULTS

In this section, we provide more visual results of *OmniFace*. Fig. 10 and Fig. 13 present our inference results, Fig. 12 and Fig. 13 show comparisons with baselines. As DynamicFace Wang et al. (2025) and ViVidFace Shao et al. (2024) do not have publicly available source codes, we extract some cases from their websites for comparison. The design of our SyncID-Pipe seamlessly inherits the capabilities of image face swapping, enabling high-fidelity results in diverse scenarios such as cartoon styles and under complex lighting. Moreover, the Synthetic-to-Real Curriculum learning strategy allows *OmniFace* to maintain high photorealism while preserving exceptional identity similarity. Furthermore, our Identity-Coherence Reinforcement Learning (IRL) ensures that *Omni-Face* maintains high similarity even under challenging conditions, including large poses and extreme expressions. A comparison with closed-source models further underscores the superior identity similarity achieved by *OmniFace*.

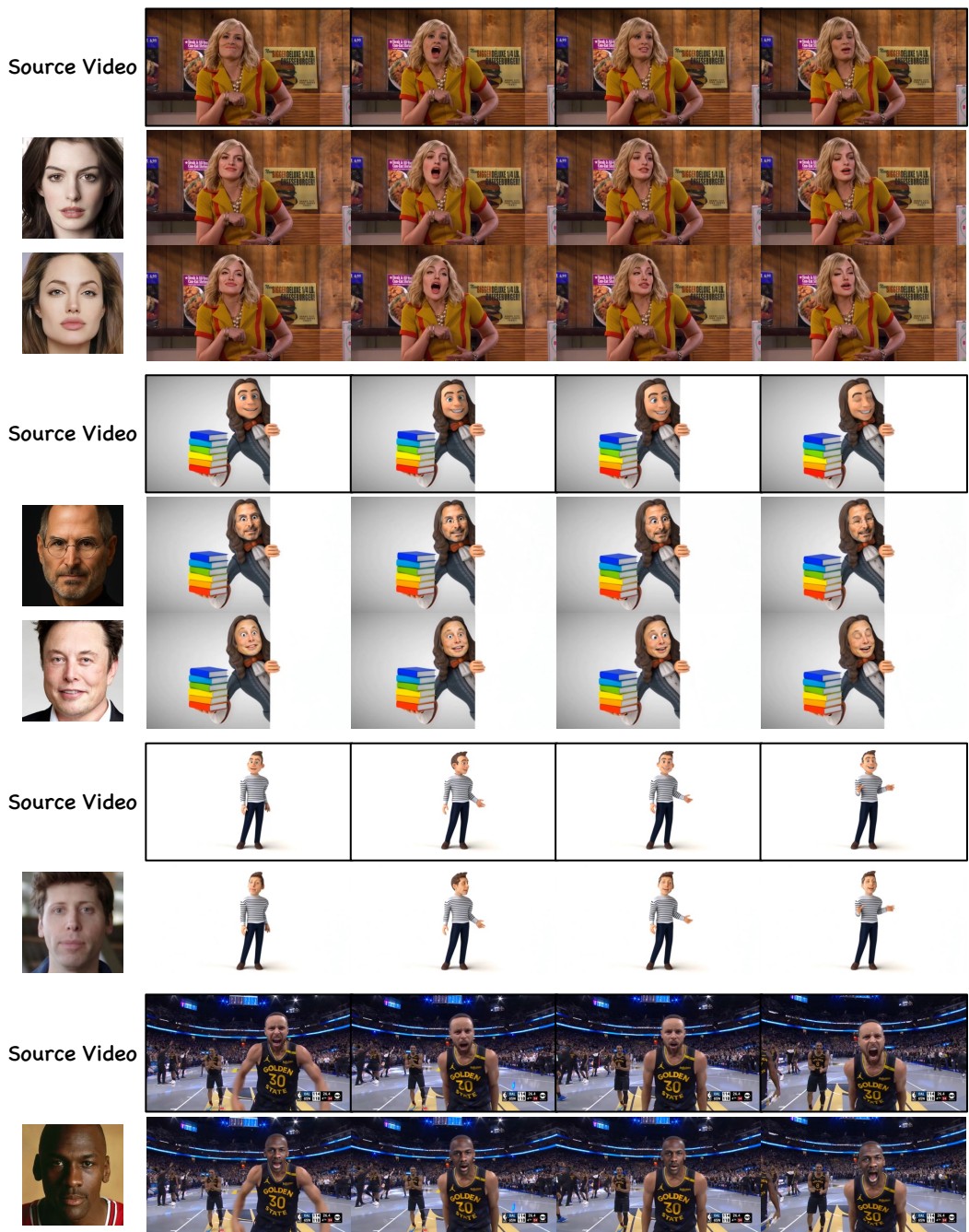

Figure 10: More **qualitative results I**.

## A.8 LIMITATIONS

Despite achieving promising results, our model currently exhibits two main limitations. Firstly, its inference speed is comparable to that of existing DiT-based video generation models, indicating room for further acceleration. Secondly, imperfect lighting preservation persists under highly complex lighting conditions. This could be addressed in future work by leveraging data augmentation based on relighting models.

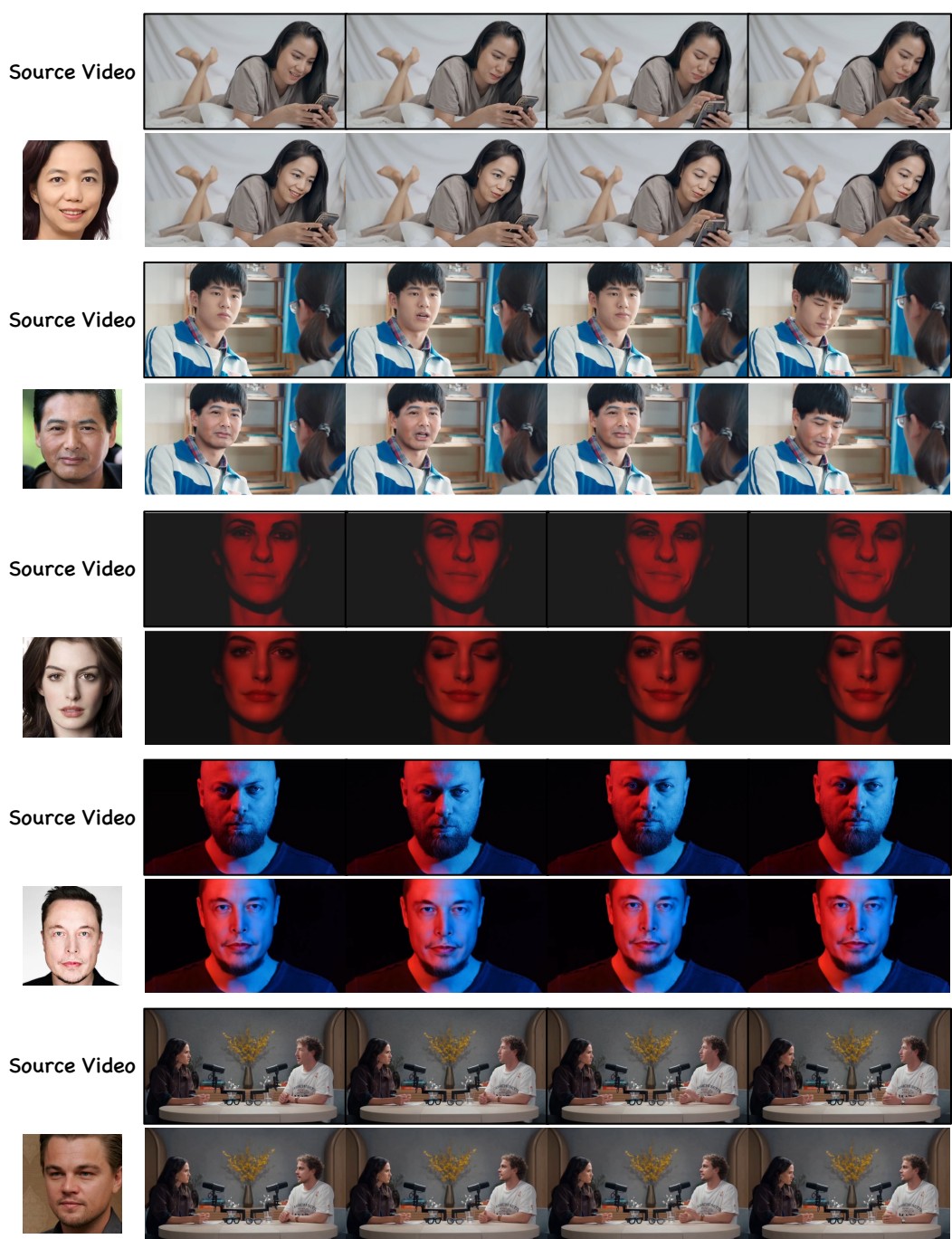

Figure 11: More **qualitative results II**.

## A.9 ETHICAL CONSIDERATIONS

OmniFace produces high-fidelity, temporally consistent face-swapped videos that could be mis-used to create non-consensual deepfakes or disinformation. To mitigate these risks we release the model under a click-through license that explicitly prohibits malicious, privacy-violating or misleading applications and require users to obtain explicit consent from any identifiable individual before publication.

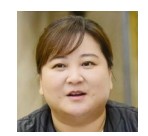

**DynamicFace**

**Ours**

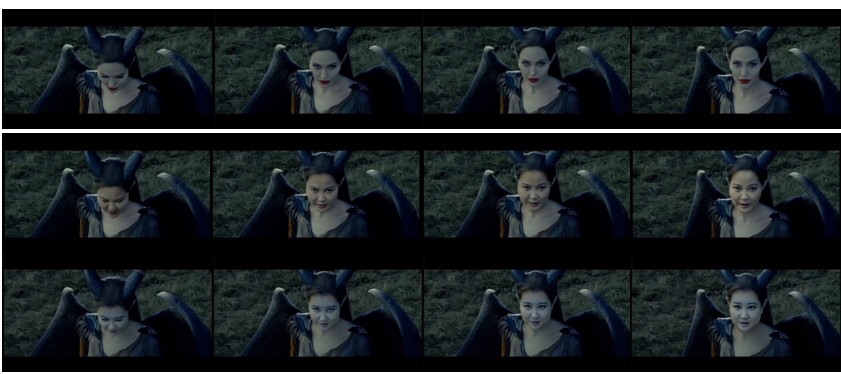

Figure 12: More **qualitative comparisons I**.

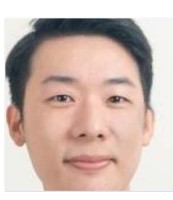

**ViVidFace**

**Ours**

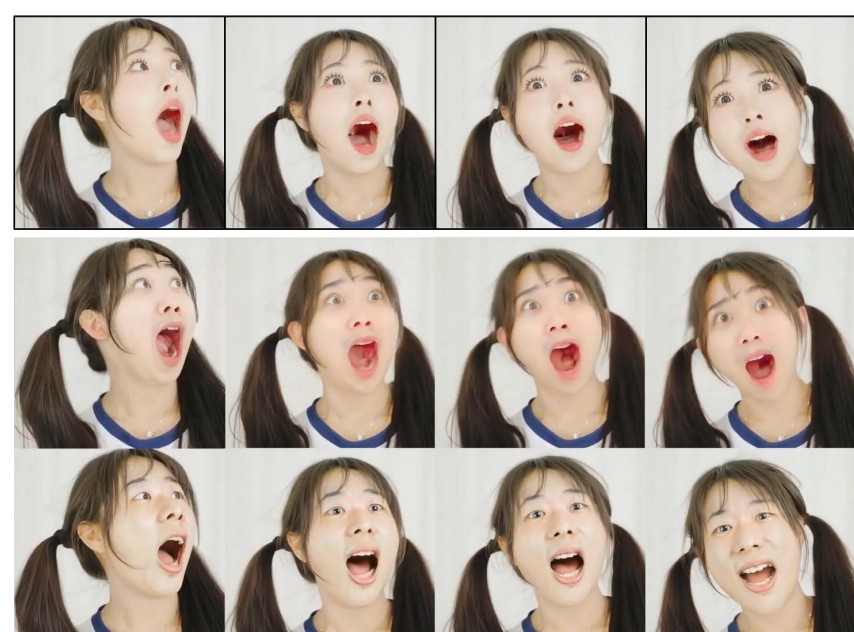

Figure 13: More **qualitative comparisons II**.

