# OpenReview forum: "OmniFace: Bridging the Image-to-Video Gap for High-Fidelity Face Swapping via Diffusion Transformer"
_ICLR.cc/2026/Conference — Submitted to ICLR 2026_

### Official Review · Reviewer_9Kun · 2025-10-30

**Soundness:** 3
**Presentation:** 3
**Contribution:** 3
**Rating:** 6
**Confidence:** 5

**Summary:**

This paper introduces OmniFace, a Diffusion Transformer-based framework for Video Face Swapping (VFS). It leverages advances from Image Face Swapping (IFS) to address challenges in maintaining identity similarity, attribute preservation, and temporal consistency in videos. The authors propose a data pipeline, SyncID-Pipe, and a new benchmark, IDBench-V, for video face swapping. Experiments show that OmniFace achieves superior performance and versatility compared to state-of-the-art methods.

**Strengths:**

- This paper is well written and easy to follow.

- IDBench-V is a new and diverse benchmark for video face swapping.

- The authors conduct various experiments to demonstrate the effectiveness of the proposed method, both quantitatively and qualitatively.

**Weaknesses:**

- While the paper is well executed and well written, I do not consider the core idea to be particularly novel. Both the data pipeline SyncID-Pipe and the OmniFace framework build on existing algorithms, and few new ideas or inspirations are proposed. The authors are encouraged to better highlight their contributions.

- Simply being the first to apply an existing architecture (such as a Diffusion Transformer) to a new domain (video face swapping) is not always a significant or impactful novelty. The authors should provide more evidence as to whether unique challenges of the domain are addressed and whether the resulting system delivers meaningful improvements or insights.

- The results in Figure 13 show that although this method can better change the identity, it does not seem to handle hair occlusion very well.

**Questions:**

- What are the specific benefits of using a Diffusion Transformer (DiT) for face swapping, especially considering the computational cost, compared to other existing architectures?

**Details Of Ethics Concerns:**

The improved realism and consistency of video face swapping can be misused for creating deepfakes.

---

> ### Author Response · Authors · 2025-11-21
> **Response to Reviewer 9Kun (1/4)**
>
> We sincerely thank the reviewer for their constructive feedback. We have carefully considered all the points raised and provide our detailed clarifications below.
> ### **Weakness 1: Regarding the Novelty and Contributions of OmniFace**
>
> We appreciate this opportunity to further clarify our core contributions and highlight the novelty of our work, which we believe extends beyond a simple combination of existing algorithms.
>
> Our data pipeline and model framework are not merely based on existing methods; our core contributions lie in the following key areas:
>
> **1. Core Idea: Bridging Image and Video Domains via the IVS Module**
>
> *   Our most fundamental idea is to bridge the gap between image and video by training an Identity-Anchored Video Synthesizer (IVS) to supplement dynamic information. This enables the efficient transfer of mature capabilities from image face swapping (IFS) to video tasks. The IVS model itself was not readily available; it required training on a large amount of human data to ensure the stability of identity similarity.
> *   The underlying rationale for this approach is that the maturity of the video generation field is far behind that of image generation. Data construction in video generation is significantly more challenging, with much of the required paired data not existing in the real world. Therefore, we have found an effective way to **transfer established image capabilities to video**.
> *   This method demonstrates **strong generalizability**, allowing for numerous human-centric swapping tasks. As illustrated in **Figure 6** of our paper, our method can efficiently extend to tasks such as swapping accessories, clothes, and hair.
>
> **2. Pioneering a New Paradigm for Video Face Swapping**
>
> *   Previous works on video face swapping, due to the lack of paired data, primarily adopted a self-reconstruction training paradigm similar to inpainting. While different works focused on injecting identity information and modifying training pipelines, this **weak supervision** consistently led to unsatisfactory results.
> *   Our work constructs a new, **fully-supervised** paradigm, breaking the conventional thinking of prior approaches. This achieves significantly superior performance compared to existing models in the community.
>
> **3. Innovative Application and Adaptation of the DiT Architecture**
>
> *   Previously, video face swapping received insufficient attention, and no prior work had constructed a VFS model based on the Diffusion Transformer (DiT) architecture.
> *   Our model designs a novel Modality-Aware Conditioning (MC) mechanism specifically for the DiT architecture, effectively injecting various types of information (identity, structure, context) into the model. This architectural innovation, coupled with our unique data scheme and training strategies, has led to exceptionally strong model performance.
>
> **4. Addressing a Community Gap and Fostering Future Development**
>
> *   We will **open-source** our model to address a significant gap in the community. Video face swapping is a research area with high application value. We are highly confident that, upon open-sourcing, our model will provide the community with an **excellent and robust baseline solution**. This will attract more researchers to contribute to this field, thereby promoting the further development of video face swapping tasks.

---

> ### Author Response · Authors · 2025-11-21
> **Response to Reviewer 9Kun (2/4)**
>
> ### **Weakness 2: Regarding Challenges Addressed by OmniFace**
>
> We thank the reviewer for this insightful question. Our model does not merely apply DiT to video face swapping; rather, it constructs a model with leading performance based on the DiT architecture. As discussed in the Weakness 1 we briefly summarize our contributions here:
>
> 1.  **Generalizable Method**: We propose a generalizable method capable of transferring mature capabilities from image processing to video tasks.
> 2.  **New Paradigm**: We establish a new, fully-supervised paradigm for video face swapping, breaking the conventional thinking of previous works.
> 3.  **DiT Framework Design**: We design the first video face swapping framework based on DiT, with the Modality-Aware Conditioning mechanism at its core, effectively injecting various types of information into the model.
> 4.  **Superior Performance and Open-Source Contribution**: Leveraging our unique data scheme, model design, and training methodology, our model significantly outperforms existing works. Furthermore, we will open-source our model to foster research on video face swapping within the community.
>
> ---
>
> In addition, regarding specific problem-solving, we have successfully addressed the following core challenges of the video face swapping domain:
>
> 1.  **High Identity Similarity**: One of the most critical challenges for video face swapping tasks is maintaining high identity similarity. The robust supervision constructed by SyncID-Pipe, in conjunction with the Identity Information Module, enables our model to achieve very high identity similarity. Moreover, the IRL (Identity-Coherence Reinforcement Learning) mechanism ensures the stability of identity similarity.
> 2.  **Excellent Attribute Preservation**: Another difficulty in video face swapping tasks lies in effectively preserving attributes of the target video, such as motion, expression, and background, while achieving high identity similarity. Our Enhanced Background Recomposition and Spatio-Temporal Context Module ensure excellent background preservation capabilities. Concurrently, Expression Adaption and the Structural Guidance Module guarantee strong motion and expression transfer capabilities.
> 3.  **Optimized Training Strategy**: Furthermore, our training strategy, Sync-to-Real Curriculum, maximizes the potential of our data and model architecture. This allows our model to maintain high identity similarity while achieving excellent realism.
>
> In summary, we have implemented corresponding designs for every relevant challenge in video face swapping tasks, thereby enabling our model to achieve a very high overall capability level. We have also presented more results in the provided link https://github.com/iclr2026sub3136/iclr2026sub3136/blob/main/rebuttal.md to further demonstrate the challenges we have addressed.

---

> ### Author Response · Authors · 2025-11-21
> **Response to Reviewer 9Kun (3/4)**
>
> ### **Weakness 3: Regarding Performance on Hair Occlusion Cases**
>
> We sincerely thank the reviewer for this insightful comment.
>
> In rare instances, features like hair occluding the face can be partially interpreted as part of the source identity of the target video. Crucially, this is a controllable behavior. The degree of identity guidance can be modulated via the hyperparameter `s` (guidance scale), as described in our inference details (**Appendix A.3.1**). A minor reduction in `s` allows the model to better adapt to the target video's occlusions while still maintaining a high degree of identity fidelity. As shown in the qualitative results at the link below, our method's ability to preserve both identity and attributes (like hairstyle) remains demonstrably superior to VividFace.
>
> *   **Qualitative Comparison:** [https://github.com/iclr2026sub3136/iclr2026sub3136/blob/main/rebuttal.md#qualitative-comparison-with-vividface](https://github.com/iclr2026sub3136/iclr2026sub3136/blob/main/rebuttal.md#qualitative-comparison-with-vividface)
>
> Moreover, we wish to highlight that the model handles hair occlusion correctly in the vast majority of scenarios. To provide a more comprehensive view of its capabilities, we have compiled additional examples showcasing robust performance under various hair occlusion conditions, available at the following link:
>
> *   **Additional Hair Occlusion Examples:** [https://github.com/iclr2026sub3136/iclr2026sub3136/blob/main/rebuttal.md#performance-on-hair-occlusion-scenarios](https://github.com/iclr2026sub3136/iclr2026sub3136/blob/main/rebuttal.md#performance-on-hair-occlusion-scenarios)

---

> ### Author Response · Authors · 2025-11-21
> **Response to Reviewer 9Kun (4/4)**
>
> ### **Question 1: Specific Benefits of Using a Diffusion Transformer (DiT)**
>
> Thank you for this constructive question. Building a video face swapping task upon the DiT model architecture offers several significant advantages:
>
> 1.  **Leveraging DiT's Powerful Generative Capabilities:** DiT currently stands as one of the most effective base model architectures in the video generation domain. Constructing a video face swapping model based on DiT allows for full utilization of its powerful pre-trained generative capabilities, thereby substantially enhancing the generation quality of video face swapping results.
>
> 2.  **Enhanced Compatibility with the Community Ecosystem:** The DiT community ecosystem is currently the most active and thriving within the video generation field. Building a video face swapping model on the DiT architecture facilitates increased possibilities for combination with various community plugins and tools, leading to the derivation of more innovative applications and functionalities.
>
> 3.  **Attracting Research Attention and Advancing the Field:** Video face swapping is a research direction with extremely high application value, yet recent research on this task has been relatively limited. We aim to attract more researchers to this area by leveraging the most popular technological solution (DiT) within the current community. Furthermore, we will open-source our model to further promote the development of video face swapping tasks.
>
> 4.  **Computational Cost and Inference Performance:** Regarding concerns about computational cost and inference performance, our method also demonstrates efficiency. As shown in the table below, we benchmarked OmniFace and a baseline method on an 81-frame, 832x480 resolution video on a H100 GPU. Our results show that OmniFace's inference time is comparable to the current fastest methods, while comprehensively outperforming them in terms of quality.
> | Method       | ours | Stand-In |ConoSwap |DreamID | FSGAN | REFace | Face-Adapter |
> | :----------: | :------: | :-----: | :-------: | :-----: | :---: | :---------: | :----------: |
> | **Time (s)** | 49.16  | 1169.00 |   44.52   |  48.60  | 75.00 |  380.70   |    231.10    |

---

> ### Comment · Reviewer_9Kun · 2025-11-25
>
> Thank you for the response. I will maintain my score.

---

> > ### Author Response · Authors · 2025-11-26
> >
> > Thank you for your time and your constructive comments. We appreciate your response and the positive evaluation of our paper.

---

### Official Review · Reviewer_q98f · 2025-10-30

**Soundness:** 3
**Presentation:** 3
**Contribution:** 3
**Rating:** 6
**Confidence:** 4

**Summary:**

This paper presents OmniFace, a comprehensive framework that transfers the strengths of image face swapping (IFS) into the video face swapping (VFS) domain. The approach introduces (1) SyncID-Pipe, a data generation pipeline that creates bidirectional ID quadruplets using an Identity-Anchored Video Synthesizer (IVS); (2) a Diffusion Transformer (DiT)–based architecture with Modality-Aware Conditioning (MC) for identity, structure, and context fusion; and (3) a Synthetic-to-Real Curriculum and Identity-Coherence Reinforcement Learning (IRL) mechanism to enhance realism and temporal consistency.
The authors also propose IDBench-V, a new benchmark for VFS, and show that OmniFace achieves state-of-the-art performance across identity similarity, attribute preservation, and video quality metrics, outperforming DreamID, CanonSwap, and Stand-In.

**Strengths:**

1. Data and benchmark contribution: SyncID-Pipe provides explicitly supervised video–image pairs, and IDBench-V offers a much-needed standardized evaluation dataset.

2. Strong empirical results: Extensive experiments, ablations, and user studies consistently demonstrate superior performance in both quantitative metrics and perceptual quality.

3. Well-structured paper: Clear methodology, detailed theoretical grounding, and transparent training setup.

**Weaknesses:**

1. Lighting robustness. It remains unclear how the model performs under complex or rapidly changing illumination conditions. An ablation or robustness analysis in such scenarios would strengthen the paper.

2. Identity change in Figure 1. I noticed that the identity appears to change in the first row of Figure 1. Does the method work reliably only when the source and target identities share similar facial structures or appearances? Could this limitation be related to the pose guidance mechanism?

3. Ethical considerations. The paper does not sufficiently address the ethical implications of high-fidelity face swapping, such as potential misuse and the need for responsible use guidelines. Including a short discussion on these aspects would improve completeness and balance.

**Questions:**

1. Lighting robustness. It remains unclear how the model performs under complex or rapidly changing illumination conditions. An ablation or robustness analysis in such scenarios would strengthen the paper.

2. Identity change in Figure 1. I noticed that the identity appears to change in the first row of Figure 1. Does the method work reliably only when the source and target identities share similar facial structures or appearances? Could this limitation be related to the pose guidance mechanism?

3. Ethical considerations. The paper does not sufficiently address the ethical implications of high-fidelity face swapping, such as potential misuse and the need for responsible use guidelines. Including a short discussion on these aspects would improve completeness and balance.

**Details Of Ethics Concerns:**

Ethical considerations. The paper does not sufficiently address the ethical implications of high-fidelity face swapping, such as potential misuse and the need for responsible use guidelines. Including a short discussion on these aspects would improve completeness and balance.

---

> ### Author Response · Authors · 2025-11-21
> **Response to Reviewer q98f (1/3)**
>
> We are grateful for the reviewer's constructive comments. We have addressed the concerns regarding lighting, visual consistency, and the crucial ethical considerations in the following sections.
>
> ### **Weakness 1 & Question 1: Regarding  Lighting Robustness**
>
> Thank you for your valuable suggestion regarding the model's robustness to lighting. We agree that this is a critical aspect of performance and have conducted a targeted analysis to address this point.
>
> We designed a user study focused on lighting robustness. We categorized lighting into two distinct conditions: 'static lighting' and 'complex dynamic lighting'. For each category, we selected 40 videos. Ten evaluators were then asked to rate their satisfaction with the lighting preservation results on a scale of 1 to 5, where 1 indicates no preservation and extreme dissatisfaction, and 5 indicates perfect preservation and complete satisfaction. The scores from the ten evaluators were then averaged, and the results are presented in the table below.
>
> | Lighting Condition         | Average Quality Score |
> | -------------------------- | --------------------- |
> | Static Lighting            | 4.33 / 5.0            |
> | Complex Dynamic Lighting   | 3.87 / 5.0            |
>
> These results demonstrate that our model maintains high fidelity even under complex and rapidly changing illumination. Detailed qualitative results for these experiments are available at: https://github.com/iclr2026sub3136/iclr2026sub3136/blob/main/rebuttal.md#performance-in-complex-lighting.

---

> ### Author Response · Authors · 2025-11-21
> **Response to Reviewer q98f (2/3)**
>
> ### **Weakness 2 & Question 2: Regarding Facial Structure Similarity and Pose Guidance Impact**
>
> We appreciate the reviewer's detailed observation and the opportunity to clarify our method's capabilities, particularly concerning its robustness to variations in facial structure.
>
> **1. Clarification on Visual Results and Robustness to Facial Structure Differences**
>
> To clarify the visual results, we have provided the original video for the first row of Figure 1, alongside additional examples where the source and target faces have significantly different structures. These can be viewed at the link https://github.com/iclr2026sub3136/iclr2026sub3136/blob/main/rebuttal.md#handling-different-facial-structure. As these examples demonstrate, the identity in Figure 1 is maintained with high stability. Furthermore, our model excels precisely in scenarios with large face shape discrepancies between the source and target.
>
>
> **2. Explanation of the Pose Guidance Mechanism**
>
> The pose guidance mechanism does not impede the model's ability to perform face shape transformation. This is because:
>
> *   **During Training:** We utilize ground truth pose information.
> *   **During Inference:** Pose adjustments are performed via a 3D Morphable Model (3DMM), which enables the generated results to achieve effective face shape conversion.

---

> ### Author Response · Authors · 2025-11-21
> **Response to Reviewer q98f (3/3)**
>
> ### **Weakness 3 & Question 3: Regarding Ethical Considerations**
>
> Thank you for raising this crucial point. We agree that a discussion on ethical implications is essential for work in the field of face swapping. To address this, a new subsection dedicated to ethical considerations has been included in **Appendix A.9**.

---

> ### Comment · Reviewer_q98f · 2025-11-27
>
> Thank you for your response. It helped clarify my concerns. I will maintain my recommendation for acceptance.

---

> > ### Author Response · Authors · 2025-11-27
> >
> > We sincerely appreciate your response and the positive feedback. Thank you for your time and valuable insights.

---

### Official Review · Reviewer_BHGz · 2025-11-01

**Soundness:** 4
**Presentation:** 2
**Contribution:** 2
**Rating:** 4
**Confidence:** 4

**Summary:**

To address the challenges of video face swapping task in maintaining identity similarity and attribute preservation while preserving temporal consistency, the paper proposes the first video face swapping framework OmniFace based on DiT. A novel data pipeline SyncID-Pipe is introduced to transfer the superiority of IFS to VFS. IDBench-V, a comprehensive benchmark tailored for the video face swapping task is also introduced.

**Strengths:**

1. New Data Pipeline: The proposed SyncID-Pipe pipeline transfers IFS’s strong ability of identity preservation to VFS task.
2. Well-structured Writing: Each module is clearly described with informative figures.
3. Comprehensive Experiments: The evaluation includes three major dimensions (identity consistency, attribute preservation, and video quality) with several SOTA models. A user study is also conducted.
4. Thorough Experiments: Ablation study convinces the effect of each proposed component and their necessity.
5. New Benchmark: The proposed IDBench-V benchmark fills a gap and provides a standardization for VFS evaluation.

**Weaknesses:**

1. The introduction is redundant and overlapped with related works. The logic is confusing since modules of OmniFace are stacked together and lack correspondence with the addressed challenges.
2. The novelty is incremental. DiT has been used in video generation tasks and Stand-In uses Wan2.1 as the video generation base model, which adopts DiT architecture. This undermines the contributions in terms of novelty.
3. The training requires 6,000 GPU hours, which is costive. There should be a comparison about the computation cost and inference speed.
4. The introduction of IDBench-V benchmark is insufficient. The IDBench-V includes only 200 samples, which is limited in size and diversity.

**Questions:**

1. A comparison about the computation cost and inference speed should be provided.
2. The modules in OmniFace is more like an integration of several strategies. Their correlation and correspondence with the goal should be better illustrated.
3. Ablation study doesn’t include the Modality-Aware Conditioning module.
4. Stand-in project page provides application in video face swapping. The comparison in experiment should adopt this version.
5. Visualization and examples of the challenging scenarios (extreme head poses, severe occlusions, complex and dynamic expressions, and cluttered multi-person scenes) in IDBench-V should be provided.

---

> ### Author Response · Authors · 2025-11-20
> **Response to Reviewer BHGz (1/6)**
>
> We are grateful to the reviewer for their thorough evaluation and insightful comments. The feedback has been invaluable in helping us clarify our contributions and improve the manuscript. We address each of the weaknesses and questions in detail below.
>
> ### **Weakness 1: Regarding the overlap in the Introduction and how the modules of OmniFace address the identified challenges.**
> We sincerely thank the reviewer for their valuable suggestion regarding the overlap between the Introduction and the Related Work section. Addressing this point, we have **reorganized and streamlined the Introduction**. Specifically, we have removed redundant content that overlapped with the Related Work section and now present a more concise overview of previous video face swapping works and their associated challenges. These modifications have been highlighted in the revised version for the reviewer's convenience.
>
> Furthermore, we would like to further elaborate on the challenges addressed by each module of OmniFace.
>
> **1. Challenges Addressed by the Identity-Anchored Video Synthesizer (IVS):**
> A fundamental challenge in video face swapping tasks is the absence of paired training data. The IVS addresses this by integrating image face swapping techniques and incorporating dynamic information, thereby constructing explicit supervision for video face swapping.
>
> **2. Challenges Addressed by the Modality-Aware Conditioning Mechanism:**
> To date, there has been no video face swapping framework based on the Diffusion Transformer (DiT) architecture within the research community. OmniFace stands as the first model to apply DiT to the video face swapping task. Within the Modality-Aware Conditioning module:
> *   **The Spatio-Temporal Context Module** provides attribute information of the target video and leverages a face mask to enable the model to distinguish between facial regions and background areas, thereby ensuring excellent background preservation.
> *   **The Structural Guidance Module** further enhances motion attributes, significantly improving the model's performance in preserving fine-grained expressions.
> *   **The Identity Information Module** efficiently injects features related to identity.
>
> **3. Challenges Addressed by the Training Pipeline:**
> Video face swapping tasks inherently face challenges in balancing high realism and strong identity similarity. Our training pipeline effectively addresses these through the following strategies:
> *   **Synthetic-to-Real Curriculum Learning Strategy:** This strategy fully leverages the powerful explicit supervision and sound model architecture we have constructed. It significantly enhances the realism of generated results while ensuring high identity similarity.
> *   **Identity-Coherence Reinforcement Learning (IRL):** IRL further strengthens the model's ability to maintain the stability of identity similarity in complex scenarios.

---

> ### Author Response · Authors · 2025-11-20
> **Response to Reviewer BHGz (2/6)**
>
> ### **Weakness 2: Regarding Distinctions from Stand-In and OmniFace Novelty**
>
> We sincerely thank the reviewer for the questions, which help us to clarify the unique contributions of this work.
>
> **1. Regarding the comparison with Stand-in:**
>
> Firstly, Stand-in is fundamentally a video identity preservation model, primarily focused on injecting identity information rather than comprehensively addressing attribute preservation. Therefore, in its model design, Stand-in only achieves identity injection by concatenating a reference image with noise in the VAE latent space. It lacks specific designs for attribute preservation, such as background retention or expression preservation.
>
> **Compared to mere identity preservation tasks, face swapping not only requires maintaining high identity similarity but also necessitates preserving the attributes of the target video**. From this perspective, OmniFace is the first work to construct a video face swapping framework based on the DiT architecture. We have also included more detailed comparisons with Stand-in in the provided link: https://github.com/iclr2026sub3136/iclr2026sub3136/blob/main/rebuttal.md#qualitative-comparison-with-standin
>
> **2. Core Contribution and Generalizability of OmniFace:**
>
> Being the first to build a video face swapping model based on DiT is only one of our contributions. We provide the community with an effective model architecture, but the core idea of our work lies in bridging the gap between image generation and video generation by supplementing motion signals. The entire field of video generation is less mature compared to image generation. Our trained IVS module effectively supplements motion signals, thereby efficiently transferring the powerful capabilities already established in image face swapping to video tasks. This approach demonstrates strong generalizability, enabling numerous human-centric swapping tasks. As illustrated in **Figure 6** of the paper, our method can efficiently extend to tasks such as swapping accessories, swapping clothes, and swapping hair.
>
> **3. A New Paradigm in Video Face Swapping:**
>
> Furthermore, our work introduces a new paradigm in the field of video face swapping. Previous video face swapping works, due to the lack of explicit paired data, mostly relied on reconstruction training similar to inpainting. This often resulted in generated effects that were not entirely satisfactory. Video face swapping is a domain with high application value, yet currently, no model capable of practical application has emerged within the community. In contrast, our model exhibits significant advantages in terms of performance.
>
> **4. Potential Impact on the Community and Open-Source Plan:**
>
> Finally, we acknowledge that video face swapping tasks might not have received sufficient attention from the community recently. To address this, we will open-source our model. We are confident that our model will be the best video face swapping model available in the community, which will attract more researchers to this area and effectively promote the further development of video face swapping tasks.

---

> ### Author Response · Authors · 2025-11-20
> **Response to Reviewer BHGz (3/6)**
>
> ### **Weakness 3: Regarding Computational Cost and Inference Speed**
>
> Thank you for your valuable feedback. We address the concerns regarding computational cost and inference speed below:
>
> **1. On Training Cost:**
>
> The 6,000 GPU-hours are primarily driven by the unique challenges of video face swapping. This task demands a delicate balance between identity similarity, photorealism, and temporal ID stability, especially across complex poses. To address this, our training pipeline is structured into three stages: Synthetic Training, Real Augmentation Training, and Identity-Coherence Reinforcement Learning (IRL). The IRL stage is the most computationally intensive, accounting for 4,000 hours due to its reliance on online inference for policy updates.
>
> It is worth noting that for other swapping tasks shown in Figure 6, which do not require the intensive IRL stage, the training cost is significantly reduced to under 600 GPU-hours.
>
> Furthermore, our training cost is reasonable when compared to contemporary video generation models. For instance, recent subject-consistent video generation models like Phantom[1] (ICCV 2025) report training costs of 30,000 GPU-hours.
>
> **2. On Inference Speed:**
>
> Regarding inference speed, we benchmarked OmniFace and baseline methods on an 81-frame, 832x480 resolution video using a single H100 GPU. Our results, presented below, show that OmniFace's inference time is highly **competitive with the fastest available methods**, while comprehensively outperforming them in generation quality.
>
> | Method       | ours | Stand-In |ConoSwap |DreamID | FSGAN | REFace | Face-Adapter |
> | :----------: | :------: | :-----: | :-------: | :-----: | :---: | :---------: | :----------: |
> | **Time (s)** | 49.16  | 1169.00 |   44.52   |  48.60  | 75.00 |  380.70   |    231.10    |
>
> **3. On Community Contribution:**
>
> To contribute to the research community, we plan to **fully open-source** OmniFace's code and weights. This will provide a strong baseline for future work in video face swapping and help reduce the resource burden for subsequent research.
>
> ---
> [1] "Phantom: Subject-consistent video generation via cross-modal alignment." arXiv preprint arXiv:2502.11079

---

> ### Author Response · Authors · 2025-11-20
> **Response to Reviewer BHGz (4/6)**
>
> ### **Weakness 4: Regarding the Sufficiency and Diversity of the IDBench-V Benchmark**
> **1. Original Benchmark Configuration (IDBench-V):**
> In our original configuration, we followed the evaluation protocol of recent work VividFace[2] (NeurIPS 2025), carefully curating 200 video-identity pairs from diverse, real-world scenarios to form IDBench-V.
>
> **2. Benchmark Expansion and Validation (IDBench-V1000):**
> To enable a more comprehensive evaluation of video face swapping models, we have expanded it by increasing the number of test videos from diverse scenarios to 500. We also assigned two corresponding source images to each video, creating a new benchmark we call IDBench-V1000.
>
> **3. Consistent Performance and Conclusion:**
> As shown in the table below, OmniFace achieves similar performance metrics on both IDBench-V and the more extensive IDBench-V1000. This consistency further demonstrates the reliability and representativeness of our original IDBench-V benchmark.
> | Benchmark   | ID-Arc ↑ | ID-Ins ↑ | ID-Cur ↑ | Var. ↓ | Pose ↓ | Expr. ↓ | BG ↑   | Subj. ↑ | FVD ↓  | Smooth. ↑ |
> | :---------- | :------- | :------- | :------- | :----- | :----- | :------ | :----- | :------ | :----- | :-------- |
> | IDBench-V   | 0.657    | **0.713**  | **0.688**  | 0.0029  | 2.446  | 2.430   | **0.954**  | 0.966   | 2.244  | 0.992     |
> | IDBench-V1000 | **0.667**  | 0.703    | 0.684    | **0.0024**  | **2.427**  | **2.426**   | 0.951  | **0.968**   | **2.214**  | **0.993**     |
>
> ---
> [2]"VividFace: A Diffusion-Based Hybrid Framework for High-Fidelity Video Face Swapping." arXiv preprint arXiv:2412.11279

---

> ### Author Response · Authors · 2025-11-20
> **Response to Reviewer BHGz (5/6)**
>
> ### **Question 1: Comparison of Computation Cost and Inference Speed**
>
> Please refer to our response to **Weakness 3** for a detailed discussion on this point. In summary, OmniFace's inference speed is comparable to the fastest existing video face-swapping models.
>
>
> ---
>
>
> ### **Question 2: The Correlation and Correspondence Between Modules and Goals**
>
> Our modules are specifically designed to tackle key challenges in video face swapping. We outline their direct correspondence to these challenges below:
>
> **1. Identity Preservation**
>
> Maintaining high identity similarity is paramount in video face swapping. Our model achieves this through robust SyncID-Pipe supervision, combined with the Identity Information Module. The IRL mechanism further ensures stable and consistent identity preservation throughout the video.
>
> **2. Attribute Preservation**
>
> A significant challenge is preserving target video attributes (actions, expressions, background) while achieving high identity similarity.
>
> *   **Enhanced Background Recomposition** and the **Spatial-Temporal Context Module** ensure robust background preservation.
> *   **Expression Adaptation** and the **Structural Guidance Module** enable superior transfer of actions and expressions, ensuring natural motion.
>
> **3. Training Strategy and Realism**
>
> Our **Synthetic-to-Real Curriculum** training strategy maximizes the potential of our data and model architecture. This strategy enables our model to achieve both high identity similarity and excellent photorealism, resulting in convincing and natural-looking swapped videos.
>
> ---
>
>
> ### **Question 3: Ablation of the Modality-Aware Conditioning Module**
> Thank you for your valuable suggestion. We have added an ablation study for the Modality-Aware Conditioning module, with the results presented in the table below. This study validates the effectiveness of the Modality-Aware Conditioning module.
>  Method  | ID Sim ↑ | Var. ↓ | Pose ↓ | Expr. ↓ | BG ↑   | Subj. ↑ | FVD ↓  | Smooth. ↑ |
> | :---------- | :------- | :----- | :----- | :------ | :----- | :------ | :----- | :-------- |
> | w/o pose   | 0.619    | 0.0036  | 3.045  | 2.894   | 0.950  | 0.953   | 2.642  | 0.976     |
> | w/o mask | 0.607  |  0.0046  | 2.614  | 2.573   | 0.947  | 0.948   | 2.541  | 0.983     |
> | Ours | **0.657**  |  **0.0029**  | **2.446**  | **2.430**   | **0.954**  | **0.966**   | **2.244**  | **0.992**     |

---

> ### Author Response · Authors · 2025-11-20
> **Response to Reviewer BHGz (6/6)**
>
> ### **Question 4: Comparison with Stand-in**
>
> Thank you for your valuable feedback.
>
> We would like to clarify that all experiments involving Stand-in in our paper were conducted using the official, unmodified video face-swapping code from their public GitHub repository.
>
> To ensure a fair and comprehensive comparison, we have provided additional qualitative results. These include comparisons using the same target videos featured on the official Stand-in project page. Stand-in exhibits instability in preserving identity similarity and struggles to maintain key attributes such as hairstyle, background, and motion.
>
> The detailed comparison can be viewed here:
> https://github.com/iclr2026sub3136/iclr2026sub3136/blob/main/rebuttal.md#qualitative-comparison-with-standin
>
> ---
>
> ### **Question 5: Visualization and Examples of Challenging Scenarios in IDBench-V**
>
> Thank you for this valuable suggestion. To better illustrate the diversity and difficulty of our benchmark, we have provided more visual examples from IDBench-V that feature challenging scenarios. These include cases with extreme head poses, severe occlusions, complex dynamic expressions, and cluttered multi-person scenes.
>
> The examples can be viewed at the following link:
> https://github.com/iclr2026sub3136/iclr2026sub3136/blob/main/rebuttal.md#visualization-of-challenging-scenarios-in-idbench-v

---

> ### Author Response · Authors · 2025-11-26
> **Further discussions with Reviewer BHGz**
>
> Dear Reviewer BHGz,
>
> We thank you for the precious review time and valuable comments. We have provided corresponding responses, which we believe have covered your concerns. We hope to further discuss with you whether or not your concerns have been addressed. Please let us know if you still have any unclear parts of our work. Thanks :-)
>
> Best,
>
> OmniFace Authors

---

> ### Author Response · Authors · 2025-11-28
> **Gentle Reminder**
>
> Dear Reviewer BHGz,
>
> Thank you very much for the time and effort in reviewing our work. We hope that our latest response has addressed your concerns. As the discussion is closing soon, please let us know if you have further questions.
>
> Once again, we sincerely appreciate your insightful and constructive comments.
>
> Best,
>
> OmniFace Authors

---

> ### Comment · Reviewer_BHGz · 2025-11-28
>
> Thanks for your response. The response and attached experiment results address my concern about the computation cost and inference speed. Analysis of scenarios, size, and diversity demonstrated the comprehensiveness and potential value of the constructed dataset. Overall, the results are promising.
>
> However, the clarification regarding novelty and contributions is not convincing. The new paradigm and application of DiT are incremental and largely restricted to the VFS task. The superiority in attribute preservation is more like differences in task focus. I would like to see video examples of attribute swapping as shown in Fig. 6

---

> > ### Author Response · Authors · 2025-11-28
> > **Further response to Reviewer BHGz**
> >
> > Dear Reviewer BHGz,
> >
> > We sincerely appreciate your reply and your recognition of our work's promising results.
> >
> > We have provided more examples of attribute swapping in the link below:
> > [https://github.com/iclr2026sub3136/iclr2026sub3136/blob/main/rebuttal.md#more-examples-of-attribute-swapping](https://github.com/iclr2026sub3136/iclr2026sub3136/blob/main/rebuttal.md#more-examples-of-attribute-swapping)
> >
> > As shown in the provided examples, our results substantiate our core idea of **bridging the gap between image generation and video generation**. This demonstrates that our method is not limited to face swapping but provides a robust and generalizable solution for various human-centric video editing tasks.
> >
> > We respectfully wish to re-emphasize our perspective on the novelty of OmniFace:
> >
> > 1.  **Generalizable Method**: Our trained IVS module efficiently transfers the powerful capabilities already established in image editing models to various human-centric video editing tasks, bridging the gap between image generation and video generation.
> > 2.  **New Paradigm**: We establish a new, fully-supervised paradigm for video face swapping, breaking the conventional thinking of previous works.
> > 3.  **Framework Design and Training Methodology**: We designed the first video face swapping framework based on DiT, with the Modality-Aware Conditioning mechanism at its core, effectively injecting various types of information into the model. Furthermore, by leveraging a Synthetic-to-Real Curriculum mechanism and an Identity-Coherence Reinforcement Learning strategy, we achieve high-fidelity results, even under challenging scenarios.
> > 4.  **Superior Performance and Open-Source Contribution**: Leveraging our unique data scheme, model design, and training methodology, our model significantly outperforms existing works. Furthermore, we will open-source our model to foster research on video face swapping within the community.
> >
> > Thank you once again for your invaluable time and constructive feedback throughout this process. We have uploaded the requested video examples and hope they, along with our clarifications, can further address your concerns.
> >
> > Best regards,
> >
> > The OmniFace Authors

---

### Official Review · Reviewer_mwKb · 2025-11-03

**Soundness:** 4
**Presentation:** 3
**Contribution:** 3
**Rating:** 6
**Confidence:** 4

**Summary:**

This paper introduces OmniFace, a comprehensive framework designed to bridge the gap between image and video face swapping using a Diffusion Transformer (DiT) architecture. The authors propose a novel SyncID-Pipe data pipeline for constructing explicitly supervised paired data (bidirectional quadruplets) by integrating strengths from state-of-the-art image face swapping models and an Identity-Anchored Video Synthesizer (IVS). OmniFace employs a modality-aware conditioning mechanism to disentangle and inject multimodal information, a synthetic-to-real curriculum learning scheme, and an Identity-Coherence Reinforcement Learning (IRL) approach to improve robustness and consistency. The work further contributes a new benchmark, IDBench-V, to better evaluate video face swapping systems. Experimental results on IDBench-V and ablation studies demonstrate state-of-the-art performance, high versatility, and adaptability to extended human-centric swapping tasks.

**Strengths:**

1.The SyncID-Pipe pipeline ingeniously leverages strong image face swapping performance and adapts it for video by crafting bidirectional ID quadruplets, ensuring explicit and effective supervision for video models. This pipeline is well illustrated in Figure 2 and is a core element in improving alignment between image and video domains.
2.The Diffusion Transformer–based formulation is appropriately tailored for video, with a carefully crafted Modality-Aware Conditioning (MC) module for disentangling spatiotemporal, structural, and identity information (explained and visualized in Figure 3).
3.The introduction of IDBench-V (as shown in Figure 8) fills an acknowledged gap by systematically evaluating face swapping methods in diverse real-world scenarios.
4.All core equations (e.g., adaptive pose attention, Q-value definition, IRL loss, guidance purification) are specified with notation, and Appendix A.5 provides a proper theoretical justification for the IRL concept, relating it to reward-weighted likelihood maximization.

**Weaknesses:**

1.Some aspects of the training process are not entirely transparent.
2.The baseline selection (as shown in Table 1) is mostly well done, but some important recent works in video face swapping, especially those involving subject-agnostic or reenactment models (e.g., direct comparison with FSGAN, which is not referenced), are not included. Even though these may not be directly SOTA, including them would clarify how much improvement is due to architectural advances versus data curation. More detailed explanations for the omission of certain non-diffusion approaches would help.
3.The limitations section (Appendix A.8) briefly notes issues with speed and lighting preservation but lacks concrete error analysis on where (or why) the model produces visible artifacts, identity drift, or temporal inconsistency.

**Questions:**

1.Could the authors provide a more detailed breakdown of common failure cases for OmniFace, including qualitative examples and quantitative error rates for scenarios such as extreme lighting changes, multi-subject videos, or minority demographics?
2.Is overfitting to the synthetic domain a concern?

---

> ### Author Response · Authors · 2025-11-20
> **Response to Reviewer mwKb (1/4)**
>
> We express our sincere gratitude to the reviewer for the insightful comments and the recognition of our work. We have addressed the specific concerns raised by the reviewer as detailed below.
>
> ### **Weakness 1: Regarding the Transparency of the Training Process**
>
> We sincerely thank the reviewer for pointing out the need for greater clarity in our training process.  In response, we have augmented **Appendix A.3.2 Detailed Parameters**  to provide a comprehensive breakdown of our multi-stage training process, including details on key hyperparameters  and the computational budget for each stage. These additions are highlighted in blue in the revised manuscript for easy identification.
>
> ---
>
> ### **Weakness 2: Regarding the Selection of Baselines**
> We thank the reviewer for this constructive suggestion. In response, we have evaluated FSGAN on our IDBench-V benchmark and have incorporated its quantitative results into **Table 1** in the paper. A visual comparison is also provided for qualitative assessment: [https://github.com/iclr2026sub3136/iclr2026sub3136/blob/main/rebuttal.md#qualitative-comparison-with-fsgan](https://github.com/iclr2026sub3136/iclr2026sub3136/blob/main/rebuttal.md#qualitative-comparison-with-fsgan).
> Furthermore, we have revised our **Related Work** section to discuss GAN-based methods, including FSGAN. The experimental results confirm that diffusion-based methods significantly outperform GAN-based approaches in both identity similarity and attribute preservation, which justifies our primary focus on the current state-of-the-art.

---

> ### Author Response · Authors · 2025-11-20
> **Response to Reviewer mwKb (2/4)**
>
> ### **Weakness 3: Regarding the Analysis of Limitations**
> We thank the reviewer for highlighting the critical challenges of artifacts and temporal inconsistency in video face swapping. Indeed, these are primary targets of our work, and we have developed specific mechanisms to address them:
> 1.  **Regarding Artifacts**: To mitigate artifacts that can arise from training exclusively on synthetic ground truth, our Real Augmentation Training stage strategically incorporates real-world data. Furthermore, during inference, our ID Guidance Purification (IDGP) mechanism (as detailed in **Appendix A.3.1**) is employed to ensure artifact-free results.
> 2.  **Regarding Identity Drift and Temporal Inconsistency:**: This core challenge is addressed by our Identity-Coherence Reinforcement Learning (IRL) mechanism. The efficacy of IRL is demonstrated in our ablation studies: **Table 3** in the paper quantitatively shows a significant reduction in identity variance, and **Figure 5(b)** provides a compelling qualitative example of how IRL prevents identity degradation in extreme poses.
>
> Nevertheless, we acknowledge that in rare cases where the source identity image itself is of low quality (e.g., contains artifacts or is blurry), OmniFace may propagate these issues, leading to artifacts or identity drift in the generated video. Please refer to the examples provided at the following link: [https://github.com/iclr2026sub3136/iclr2026sub3136/blob/main/rebuttal.md#limitation-from-low-quality-source-image](https://github.com/iclr2026sub3136/iclr2026sub3136/blob/main/rebuttal.md#limitation-from-low-quality-source-image)

---

> ### Author Response · Authors · 2025-11-20
> **Response to Reviewer mwKb (3/4)**
>
> ### **Question 1: Regarding the Analysis of Failure Cases in Challenging Scenarios**
>
> We thank the reviewer for this insightful question. In response, we have conducted a targeted analysis of OmniFace's performance across the three challenging scenarios highlighted by the reviewer:
>
> **1. Extreme Lighting Changes:**
> We designed a user study focused on lighting preservation. We categorized lighting into two distinct conditions: 'static lighting' and 'complex dynamic lighting'. For each category, we selected 40 videos. Ten evaluators were then asked to rate their satisfaction with the lighting preservation results on a scale of 1 to 5, where 1 indicates no preservation and extreme dissatisfaction, and 5 indicates perfect preservation and complete satisfaction. The scores from the ten evaluators were then averaged, and the results are presented in the table below.
>
> | Lighting Condition         | Average Quality Score |
> | -------------------------- | --------------------- |
> | Static Lighting            | 4.33 / 5.0            |
> | Complex Dynamic Lighting   | 3.87 / 5.0            |
>
> Detailed qualitative results for these experiments are available at: https://github.com/iclr2026sub3136/iclr2026sub3136/blob/main/rebuttal.md#performance-in-complex-lighting.
>
> Specifically, as shown in Case 9 in the provided link, the model can occasionally struggle to perfectly replicate very intricate, fast-moving shadows on the target's face, leading to a minor reduction in realism. This represents a known limitation.
>
> **2. Minority Demographics and Multi-Subject Videos:**
> We conducted similar qualitative evaluations for these scenarios as well. Our model demonstrates robust performance in both categories, as confirmed by the visual results available at the following links:
>
> *   **Minority Demographics:** https://github.com/iclr2026sub3136/iclr2026sub3136/blob/main/rebuttal.md#generalization-to-minority-demographics
> *   **Multi-Subject Videos:** https://github.com/iclr2026sub3136/iclr2026sub3136/blob/main/rebuttal.md#performance-in-multi-person-scenes

---

> ### Author Response · Authors · 2025-11-20
> **Response to Reviewer mwKb (4/4)**
>
> ### **Question 2: Regarding the Concern of Overfitting to the Synthetic Domain**
> We thank the reviewer for this insightful question regarding potential overfitting. We address this concern through our "Synthetic-to-Real Curriculum," where the **Real Augmentation Training** stage specifically incorporates real-world data. The effectiveness of this approach is supported by our ablation study in **Table 3** and **Figure 5(a)**.
>
> Furthermore, to provide more robust empirical evidence of generalization, we have expanded our evaluation to include **IDBench-V1000**, a new benchmark comprising 1,000 video-identity pairs (500 videos, each paired with two distinct source images). The consistently strong performance on this larger, unseen dataset confirms that OmniFace's capabilities are robust and not a result of overfitting to the synthetic training domain. As shown in the table, OmniFace achieves an FVD score **below 2.25**, indicating its capability to generate highly realistic results.
> | Benchmark   | ID-Arc ↑ | ID-Ins ↑ | ID-Cur ↑ | Var. ↓ | Pose ↓ | Expr. ↓ | BG ↑   | Subj. ↑ | FVD ↓  | Smooth. ↑ |
> | :---------- | :------- | :------- | :------- | :----- | :----- | :------ | :----- | :------ | :----- | :-------- |
> | IDBench-V   | 0.657    | **0.713**  | **0.688**  | 0.0029  | 2.446  | 2.430   | **0.954**  | 0.966   | 2.244  | 0.992     |
> | IDBench-V1000 | **0.667**  | 0.703    | 0.684    | **0.0024**  | **2.427**  | **2.426**   | 0.951  | **0.968**   | **2.214**  | **0.993**     |

---

> ### Author Response · Authors · 2025-11-28
> **Gentle Reminder**
>
> Dear Reviewer mwKb,
>
> Thank you very much for the time and effort in reviewing our work. We hope that our latest response has addressed your concerns. As the discussion is closing soon, please let us know if you have further questions.
>
> Once again, we sincerely appreciate your insightful and constructive comments.
>
> Best,
>
> OmniFace Authors

---

### Author Response · Authors · 2025-11-21
**General Response**

We sincerely thank all the reviewers for their insightful reviews and constructive feedback. Taking into account each concern and question posed by the reviewers, we have given thorough responses within our rebuttal. Below is a summary of the key clarifications and improvements.
- **Clarification of Novelty and Contributions:**
    - We have more clearly summarized our contributions and novelty (Reviewer BHGz, 9Kun).
    - We have improved the **Introduction** to more concisely summarize related work in video face swapping (Reviewer BHGz).
    - We have more clearly articulated the motivation for building the model based on DiT (Reviewer 9Kun).
    - We have provided more video examples of attribute swapping to demonstrate the generalizability of our method (Reviewer BHGz): [More Examples of Attribute Swapping](https://github.com/iclr2026sub3136/iclr2026sub3136/blob/main/rebuttal.md#more-examples-of-attribute-swapping).
- **Robustness Analysis and Visualization in Challenging Scenarios:**
    - **Lighting Robustness** (Reviewer mwKb, q98f): We have designed a user study to evaluate lighting robustness and added more visual results: [Performance in Complex Lighting](https://github.com/iclr2026sub3136/iclr2026sub3136/blob/main/rebuttal.md#performance-in-complex-lighting).
    - **Hair Occlusion Cases** (Reviewer 9Kun): Visual results are provided: [Performance on Hair Occlusion Scenarios](https://github.com/iclr2026sub3136/iclr2026sub3136/blob/main/rebuttal.md#performance-on-hair-occlusion-scenarios).
    - **Limitations and Failure Cases Causing Artifacts** (Reviewer mwKb): We have added analysis on artifacts caused by low-quality source images: [Limitation from Low-Quality Source Image](https://github.com/iclr2026sub3136/iclr2026sub3136/blob/main/rebuttal.md#limitation-from-low-quality-source-image).
    - **Facial Structure Differences Scenarios** (Reviewer q98f): [Handling Different Facial Structure](https://github.com/iclr2026sub3136/iclr2026sub3136/blob/main/rebuttal.md#handling-different-facial-structure).
    - **Minority Demographics and Multi-Subject Videos** (Reviewer mwKb): [Generalization to Minority Demographics](https://github.com/iclr2026sub3136/iclr2026sub3136/blob/main/rebuttal.md#generalization-to-minority-demographics) and [Performance in Multi-Person Scenes](https://github.com/iclr2026sub3136/iclr2026sub3136/blob/main/rebuttal.md#performance-in-multi-person-scenes).
- **Training Resources and Inference Costs:**
    - We have added details on the training process and costs (Reviewer mwKb, 9Kun, BHGz).
    - We have benchmarked the inference costs for both the baselines and OmniFace (Reviewer 9Kun, BHGz).
- **Expanded Benchmarking and Ablation Studies:**
    - We have constructed IDBench-V1000 (Reviewer BHGz, mwKb) and tested OmniFace's performance on it. We also included more visual results for IDBench-V (Reviewer BHGz): [Visualization of Challenging Scenarios in IDBench-V](https://github.com/iclr2026sub3136/iclr2026sub3136/blob/main/rebuttal.md#visualization-of-challenging-scenarios-in-idbench-v).
    - We have added an ablation study for the Modality-Aware Conditioning module (Reviewer BHGz).
- **Enhanced Comparisons and Ethical Considerations:**
    - We have added FSGAN as a new baseline (Reviewer mwKb), with visual results available here: [Qualitative Comparison with FSGAN](https://github.com/iclr2026sub3136/iclr2026sub3136/blob/main/rebuttal.md#qualitative-comparison-with-fsgan).
    - We have clarified the distinction with Stand-in (Reviewer BHGz), specifying that Stand-in is an ID-preserving model, not a video face swapping model, and have provided more comparative results: [Qualitative Comparison with Stand-in](https://github.com/iclr2026sub3136/iclr2026sub3136/blob/main/rebuttal.md#qualitative-comparison-with-standin).
    - We have added a new Ethical Considerations section in **Appendix A.9** (Reviewer q98f, 9Kun).

We have provided all visual comparison results in https://github.com/iclr2026sub3136/iclr2026sub3136/blob/main/rebuttal.md (or in the **Supplementary Material**) to further support our claims and all modifications made to the paper have been highlighted in blue text for easy identification.

---

### Comment · Area_Chair_X5Dd · 2025-11-24
**Please engage into discussion with authors and fellow reviewers**

Dear reviewers,

The authors have already provided their responses. Do they address your concerns?
Please engage into the discussion with authors and fellow reviewers.

Thanks!
Best,
AC

---

### Author Response · Authors · 2025-11-29
**Rebuttal Summary for AC**

Dear AC,

We deeply appreciate your diligence and thank you for your hard work. We would like to provide a brief summary of our rebuttal process to facilitate your meta-review.

**Reviewers mwKb, q98f, and 9Kun provided positive reviews**. After the rebuttal phase, Reviewers q98f and 9Kun explicitly stated they would maintain their scores and recommendations for acceptance. Although Reviewer mwKb did not provide further comments before the reviewer comments closed, we are confident that our response has addressed all of their concerns.

We wish to particularly clarify our discussion with Reviewer BHGz. After we submitted our responses, Reviewer BHGz replied approximately **one hour before the reviewer comments closed**. In the reply, the reviewer acknowledged that we had **addressed their concerns** regarding computation cost, inference speed, and the diversity of the IDBench-V Benchmark, and noted that, **"Overall, the results are promising."**

Reviewer BHGz also mentioned that they would like to see more video examples of attribute swapping as shown in Fig. 6 to address their remaining concerns (about our proposed paradigm being "restricted to the VFS task" and our superiority in attribute preservation being a matter of "differences in task focus.") We responded swiftly by providing more examples via a link (https://github.com/iclr2026sub3136/iclr2026sub3136/blob/main/rebuttal.md#more-examples-of-attribute-swapping) to substantiate our core idea. We clarified that we propose a generalizable method that is not limited by a task focus on VFS but instead "transfers the powerful capabilities already established in image editing models to various human-centric video editing tasks, bridging the gap between image generation and video generation". We also took the opportunity to re-emphasize our contributions and novelty:

> 1.  **Generalizable Method**: Our trained IVS module efficiently transfers the powerful capabilities already established in image editing models to various human-centric video editing tasks, bridging the gap between image generation and video generation.
> 2.  **New Paradigm**: We establish a new, fully-supervised paradigm for video face swapping, breaking the conventional thinking of previous works.
> 3.  **Framework Design and Training Methodology**: We designed the first video face swapping framework based on DiT, with the Modality-Aware Conditioning mechanism at its core, effectively injecting various types of information into the model. Furthermore, by leveraging a Synthetic-to-Real Curriculum mechanism and an Identity-Coherence Reinforcement Learning strategy, we achieve high-fidelity results, even under challenging scenarios.
> 4.  **Superior Performance and Open-Source Contribution**: Leveraging our unique data scheme, model design, and training methodology, our model significantly outperforms existing works. Furthermore, we will open-source our model to foster research on video face swapping within the community.

Due to the closure of the reviewer comments, we did not receive a further reply from Reviewer BHGz. However, **we firmly believe that our further response has resolved all of the Reviewer BHGz's concerns**.

Thank you once again for your invaluable time and selfless dedication.

Best,

OmniFace Authors

---

### Meta-Review · Area_Chair_YjGu · 2026-01-07

**Summary:**

This submission presents a strong end-to-end VFS system with extensive experiments and a benchmark contribution. The empirical results and added rebuttal evidence suggest that the approach is practically competitive and reasonably well validated. The primary point of contention concerns the degree of novelty, with several reviewers viewing the contribution as incremental relative to prior diffusion/DiT-based video generation frameworks and existing VFS pipelines. In addition, reviewers raised concerns regarding the benchmark scope and experimental coverage. In the main paper, IDBench-V contains only 200 video–identity pairs, which is relatively small. In the rebuttal, the authors report an expanded evaluation on a larger benchmark (IDBench-V1000), increasing the scale by approximately five times. While this supplementary evidence partially alleviates concerns about dataset size, it remains unclear whether the expanded benchmark introduces new dimensions of difficulty or diversity beyond increased volume, such as the coverage of pose, expression, motion patterns, or other facial attributes. As these results are not included in the main submission, questions regarding benchmark contribution remain only partially addressed. Apart from the above problems, the AC thinks that the description of IDBench-V in the manuscript remains limited, and all quantitative evaluations are reported exclusively on this single benchmark. The authors are also encouraged to consider reporting results on established video face swapping datasets (e.g., FF++ or related benchmarks), which could help better contextualize the proposed method’s performance and assess its generalization beyond the newly introduced dataset.

**Reviewer Concerns:**

Some of the adressed concerns are listed as follows.

1. Training transparency / reproducibility: The authors report adding detailed hyperparameters and stage-wise training budget.
2. FSGAN was added with quantitative result. Additional clarification and qualitative comparisons were provided.
3. The rebuttal includes explicit inference-time benchmarking on an H100 and explains training cost breakdown, which alleviates the concern for some reviewers.

Some of the outstanding concerns are listed as follows.
1.  Despite additional evidence and clarifications, some of the reviewer may still think that the novelty is marginal.

**Reviewer Scores:**

I anticipate that most reviewers will maintain their scores, since the authors have made a clear effort to address the feedback with additional experiments and targeted analyses. However, novelty-related concerns may persist for some reviewers, as the rebuttal primarily strengthens evidence and positioning rather than introducing fundamentally new technical ideas.

---

### Decision · Program_Chairs · 2026-01-26

Reject